# Barriers to Counterfactual Credit Attribution for Autoregressive Models

**Aloni Cohen** [1]    **Chenhao Zhang** [2]

## Abstract

Generative AI disrupts the practice of giving credit to work that came before. Ideally, a generative model would give credit to any work on which its output depends in a significant way. *Counterfactual credit attribution (CCA)* is a technical condition formalizing this goal—a relaxation of differential privacy—recently introduced by Livni, Moran, Nissim, and Pabbaraju (2024) who studied it in the PAC learning setting. We initiate the study of CCA generative models. Specifically, we consider autoregressive models giving credit to a deployment-time dataset (e.g., a RAG database). We uncover barriers to two natural approaches to CCA autoregressive models. First, we show that imposing CCA on the underlying next-token predictor does not guarantee that the model is CCA: CCA does not compose autoregressively (unlike DP). Second, we consider a different approach to building CCA models which we call *retrofitting*. Retrofitting takes a model that does not attribute credit, and adds credit onto it. We prove a lower bound for CCA retrofitting under a weak optimality requirement. Given black-box access to the starting model, retrofitting requires query complexity exponential in the length of the model's outputs.

## 1. Introduction

People must give credit where credit is due. In academic writing, failing to attribute sources is professional misconduct at best, plagiarism at worst. Intellectual property law (e.g., copyright) raises the stakes: one must not only identify prior art, but also ensure its use is permitted. Thankfully, it isn't hard for a human creator to give appropriate credit to their influences. The system mostly works.

Generative AI disrupts the practice of giving credit to work that came before. When genAI is used in the process of creation, the link between inputs and outputs—between old and new—is obscured. The genAI-assisted creator is unable to credit her influences, undermining the economic, professional, reputational, and legal incentives for good work in the first place.

GenAI tools should also give credit where credit is due.

**Counterfactual Credit Attribution (CCA)**    Credit attribution was studied by Livni, Moran, Nissim, and Pabbaraju (Livni et al., 2024). They consider algorithms $M(S)$ that return, in addition to the primary output $y$, a *credit set* $C \subseteq S$ of inputs to which credit is attributed.[1] Livni et al. (2024) ask what properties such algorithms should satisfy. They propose *Counterfactual Credit Attribution (CCA)*, described below, which requires crediting any input on which the output significantly depends. They study the power of CCA learning algorithms[2], leaving the study of CCA generative models for future work.

CCA is a relaxation of differential privacy (DP) that only requires stability with respect to non-credited inputs. Roughly, an algorithm $A$ satisfies $(\varepsilon, \delta)$-CCA if the following two distributions are $(\varepsilon, \delta)$-close for all datasets $S$ and documents $s_i \in S$: (1) $A(S)$ conditioned on $s_i \notin C$; and (2) $A(S \setminus \{s_i\})$. The former is the *conditional distribution* where $s_i$ isn't credited and is denoted $A^{-i}(S)$. The latter is the *counterfactual distribution* where $s_i$ is excluded from the input and is denoted $A_{-i}(S)$.

The number of items credited $|C|$ is important to consider. In many settings each credited document $s \in C$ increases the end user's costs (e.g., requiring due diligence or licensing). The smaller $|C|$ is, the stricter CCA's requirement and the stronger its guarantees. At one extreme, every algorithm is CCA if $C$ always equals $S$. At the other, CCA collapses to DP if $C$ is always empty.

[1]University of Chicago [2]Northwestern University. Correspondence to: Aloni Cohen <aloni@g.uchicago.edu>, Chenhao Zhang <chenhao.zhang.rea@u.northwestern.edu>.

*Proceedings of the 43$^{rd}$ International Conference on Machine Learning*, Seoul, South Korea. PMLR 306, 2026. Copyright 2026 by the author(s).

---

[1]This treats credit as binary and individualized, as in academic writing and IP law. In this way, credit attribution asks for much less than a Shapley value, say, and should be a simpler problem.

[2]For example, they prove that $(\varepsilon = 0, \delta = 0)$-CCA algorithms can PAC learn any concept class with finite VC dimension $d$ while crediting only $|C| = O(d(\log d + \log \log d))$ items and without much increase in sample complexity.

## 1.1. Our contributions

We initiate the study of CCA generative models, focusing specifically on autoregressive language models, and uncover barriers to two natural approaches to building generative models that attribute credit.

We impose the CCA condition on the trained generative model with respect to a deployment-time dataset.[3] For example, a database used for retrieval-augmented generation or examples for in-context learning. That is, we consider algorithms of the form $A : (S, x) \mapsto (y, C)$ and require that $A^{-i}(S, x)$ be $(\varepsilon, \delta)$-close to $A_{-i}(S, x)$.

**Non-composition of autoregressive CCA next-token prediction** An attractive approach to building CCA generative models is to reduce the problem to CCA next-token prediction. Namely, design a next-token predictor $M$ that attributes credit for individual tokens and run $M$ autoregressively to generate a sequence of tokens $y_1 y_2 \ldots$. For the final credit set, return the union of the credit sets $C_i$ for each token $y_i$. This follows the usual approach for DP language models and language models generally.

For this to work, CCA has to *compose* under autoregressive generation. That is, the generative model $G^M$ resulting from autoregressively sampling from an $(\varepsilon, \delta)$-CCA next-token predictor $M$ should be an $(\varepsilon', \delta')$-CCA for some $\varepsilon'$ and $\delta'$.

*Q1: Does CCA compose autoregressively?*

The answer is no.

**Theorem** (4.2, informal). *For all $\varepsilon > 0$, $0 \leq \delta < 1$, there exists a $(0, 0)$-CCA next-token predictor $M$ such that the induced generative model $G^M$ is not $(\varepsilon, \delta)$-CCA.*

The proof is a simple counterexample. We also prove a more general CCA composition lower bound, of which the preceding counterexample is a special case.

**Theorem** (4.3, informal). *Suppose $M$ is $(\varepsilon, 0)$-CCA and $G^M$ is $(\varepsilon', 0)$-CCA. Then*

$$\varepsilon' \geq \max_{x, y, S, s_i} \big( f(x, y, S, s_i) - |y| \cdot \varepsilon \big),$$

*where $x$ is a prompt, $y$ is a generated output, $S$ is a dataset, $s_i \in S$ is a document, and $f$ is a function independent of $\varepsilon$.*

Counterintuitively smaller $\varepsilon$ makes the lower bound on $\varepsilon'$ bigger! One would expect that as a next-token predictor gets better at attributing credit, its rollout would too. (Note however that $f$ is a function of the model which itself depends on $\varepsilon$, complicating this simple interpretation.)

**Infeasibility of CCA Retrofitting** A different approach to building CCA generative models is what we call *CCA retrofitting*: turn a model that does not attribute credit into one that satisfies CCA. We consider black-box retrofitting algorithms that act as a wrapper around a given non-crediting model $M$: they take prompts as input, query $M$ as needed, and generate outputs along with credit sets.[4]

*Q2: Is CCA retrofitting computationally feasible?*

More formally, an algorithm $A$ solves the CCA Retrofitting problem if it takes a non-crediting next-token predictor $M$ as input and produces a crediting model $\widetilde{G^M}$ that is $(\varepsilon, \delta)$-CCA. To be meaningful, we require two things. First, the credit set must not be much bigger than required. Second, $\widetilde{G^M}$ must preserve model behavior in the following sense. Let $G^M$ be the generative model induced by $M$. We require $\widetilde{G^M}$ *augments* $G^M$: the marginal distributions over generated text are equal.[5]

Unfortunately, we prove that CCA Retrofitting is computationally infeasible. For $\delta = 0$, any algorithm for CCA Retrofitting must make exponentially-many queries to the original model $M$ in the worst case. This holds as long as the augmented CCA model credits documents with probability within $\alpha < 1/2$ of an optimum solution (for a very weak notion of optimality).

**Theorem** (5.5, informal). *There exists a family of models $\mathcal{M}$ generating length $\ell$ strings such that for all oracle algorithms $A$ for the $(\varepsilon, 0)$-CCA Retrofitting problem, there exists model $M \in \mathcal{M}$ and prompt $x$ such that one of the following holds:*

*(1) $A^M(S, x)$ makes $\widetilde{\Omega}(2^\ell)$ queries to $M$ in the worst case.*

*(2) $A^M(S, x)$ credits $s \in S$ with probability $\geq \mathsf{OPT} + 1/2$.*

## 1.2. Other related work

Our paper builds on the work of Livni et al. (2024) which introduced Counterfactual Credit Attribution. We briefly highlight two other broadly related lines of work. First is work on DP LLMs, discussed further in §3. We take inspiration from the common approach to building DP LLMs of composing a private next-token predictor to get end-to-end privacy guarantees (Majmudar et al., 2022; Amin et al., 2024). Second is the large body of work on *data attribution* (Deng et al., 2025). For example, many approaches are based on Shapley values (Ghorbani & Zou, 2019). CCA is coarser—it treats credit as binary—and is formulated

---

[3]In contrast, Livni et al. (2024) require the training algorithm be CCA with respect to the training dataset. We discuss these alternatives in §3.

[4]A similar approach was recently used to achieve a different copyright-motivated relaxation of DP called near access-freeness (Vyas et al., 2023), though it isn't possible for DP itself.

[5]Dropping this requirement would trivialize the problem, say, by ignoring $S$ altogether. We leave open relaxing exact augmentation to approximate augmentation.

differently. With some exceptions (Nematov et al., 2025), most work in this direction is concerned with training data attribution, rather than attribution for a deployment-time dataset.

## 2. Preliminaries

**Notation**  Let $\mathcal{X}$ be a token vocabulary, the set of symbols from which model inputs/prompts and outputs/generations are constructed. We assume $\mathcal{X}$ is finite and there exists a special end-of-sequence token $\perp$. The set of finite sequences of tokens is denoted $\mathcal{X}^*$, and $\lambda \in \mathcal{X}^*$ is the empty sequence. We usually denote tokens by $x, y \in \mathcal{X}$, and write sequences of tokens as $\mathbf{x} \in \mathcal{X}^*$. For $\mathbf{x}, \mathbf{x}' \in \mathcal{X}^*$, $\mathbf{x} \parallel \mathbf{x}'$ denotes their concatenation. $\mathbf{x} \sqsubseteq \mathbf{x}'$ means $\mathbf{x}$ is a prefix of $\mathbf{x}'$.

A dataset $S \subseteq \mathcal{S}$ is a set of *documents* $s$ from some data universe $\mathcal{S}$. The set of possible datasets $S$ is denoted $2^{\mathcal{S}}$.

For distributions $P$ and $Q$ defined over the same space and constants $\varepsilon, \delta \geq 0$, we say $P \approx_{\varepsilon, \delta} Q$ if for all events $E$ we have $P(E) \leq e^{\varepsilon} \cdot Q(E) + \delta$ and $Q(E) \leq e^{\varepsilon} \cdot Q(E) + \delta$.

For a randomized algorithm $A : \mathcal{W} \to \Omega$, we denote by $A(w)$ the distribution over $\Omega$ generated by running $A$ on input $w$. For an outcome $o \in \Omega$, we denote by $A(o \mid w)$ the probability mass of $o$ in $A(w)$, and denote by $o \sim A(w)$ a sample from $A(w)$. We use this notation for models $M$, $G$ and their credit-attributing counterparts $\widetilde{M}, \widetilde{G}$. For $A$ as above, we allow an oracle algorithm $B^A$ to make two types of queries to $A$: (i) On input $w$, sample an outcome from the distribution $A(w)$, and (ii) On input $w, o$, compute the probability $A(o \mid w)$.

**Next-Token Prediction and Autoregressive Composition**
A next-token predictor $M : 2^{\mathcal{S}} \times \mathcal{X}^* \to \mathcal{X}$ is a randomized algorithm mapping a dataset $S$ and a prompt $\mathbf{x}_0 \in \mathcal{X}^*$ to (a distribution over) a next token: $x \sim M(S, \mathbf{x}_0)$. We assume that if $\mathbf{x}_0$ contains $\perp$, then $M(\perp \mid S, \mathbf{x}_0) = 1$.

When composed with itself autoregressively (see Algorithm 1), a next-token predictor $M$ induces a randomized generative model $G^M : 2^{\mathcal{S}} \times \mathcal{X}^* \to \mathcal{X}^*$. We call $G^M$ the *rollout* of $M$. The rollout $G^M$ iteratively samples a next token $x \sim M(S, \mathbf{x})$ by calling $M$ on the current prompt $\mathbf{x}$ and updating the prompt $\mathbf{x} = \mathbf{x} \parallel x$ until it samples the end of sequence token $\perp$. The resulting distribution over generated sequences is denoted $G^M(S, \mathbf{x}_0)$, where $\mathbf{x}_0$ is the initial prompt. By construction $\mathbf{x}_0$ is always a prefix of the output.

## 3. Counterfactual Credit Attribution

Livni et al. (2024) introduce the notion of *counterfactual credit attribution* (CCA), a relaxation of differential privacy, for data-dependent algorithms.

---

**Algorithm 1** $G^M$: rollout of next-token predictor $M$

---

**input**  dataset $S$, prompt $\mathbf{x}_0$
**output**  generated sequence $\mathbf{x}$
1:   $y \leftarrow \lambda$ {init. to empty string}
2:   $\mathbf{x} \leftarrow \mathbf{x}_0$
3:   **while** $y \neq \perp$ **do**
4:     $y \sim M(S, \mathbf{x})$ {sample next token from $M$}
5:     $\mathbf{x} \leftarrow \mathbf{x} \parallel y$
6:   **end while**
7:   **return** $\mathbf{x}$

---

**Definition 3.1** (Credit attributing algorithm). *Let $\widetilde{A} : 2^{\mathcal{S}} \to \mathcal{Y} \times 2^{\mathcal{S}}$ be a randomized algorithm which takes as input a dataset $S \subseteq \mathcal{S}$ and returns a pair $(y, C)$ for some output domain $\mathcal{Y}$. $\widetilde{A}$ is* credit-attributing *(crediting for short) if the credit set $C$ is always a subset of the input dataset $S$.*

We indicate credit attributing algorithms with a $\widetilde{\text{tilde}}$. We view $y$ as the useful output of $\widetilde{A}$—it has utility even without $C$. The credit set $C \subseteq S$ indicates to which documents $\widetilde{A}$ attributes credit for the output $y$. We use "$\widetilde{A}(S)$ credits $s_i$" to denote the event that $s_i \in C$ for $(y, C) \sim \widetilde{A}(S)$.

A credit-attributing algorithm $\widetilde{A}$ is a *counterfactual credit attributor* if the output distribution when a document is not credited is close to the counterfactual where the document had been excluded from the input dataset. Closeness is measured as in $(\varepsilon, \delta)$-differential privacy.

**Definition 3.2** (Counterfactual Credit Attribution (Livni et al., 2024)). *Let $\varepsilon, \delta \geq 0$. A (crediting) algorithm $\widetilde{A} : 2^{\mathcal{S}} \to \mathcal{Y} \times 2^{\mathcal{S}}$ is an $(\varepsilon, \delta)$-counterfactual credit attributor (CCA) if for all $S \subseteq \mathcal{S}$ and every $s_i \in S$ the following holds: either $\Pr[\widetilde{A}(S) \text{ credits } s_i] = 1$ or[6]*

$$\widetilde{A}(S^{-i}) \approx_{\varepsilon, \delta} \widetilde{A}(S_{-i}),$$

*where $\widetilde{A}(S^{-i})$ is the output distribution on the dataset $S$ conditioned on $s_i \notin C$, and $\widetilde{A}(S_{-i})$ is the output distribution on the dataset $S_{-i} = S \setminus \{s_i\}$.*

*When $\delta = 0$, we write $\varepsilon$-CCA instead of $(\varepsilon, 0)$-CCA.*

### 3.1. Applying CCA to generative models

Broadly, there are three ways to apply CCA to generative models: to the training algorithm, to the deployed model, or end-to-end. As we explain next, our work focuses on deployment-time CCA.

*Training-time CCA:* The straightforward application of Livni et al. (2024) to generative models requires the training algorithm to be CCA with respect to the training data (analogous to DP training (Abadi et al., 2016)). The semantics

---

[6]The former prevents conditioning on zero probability events.

are easy to understand: the output model parameters are credited to training data items. But that's not what we're after. We want credit to be particularized to individual generated outputs, rather than a fixed credit set for the model as a whole.

*Deployment-time CCA:* We apply CCA to the deployed generative model. We model the generation algorithm as having access to a deployment-time dataset other than the pretraining or fine-tuning dataset. For instance, a database used for retrieval-augmented generation (RAG) or examples for in-context learning included in a system prompt (for DP analogues see Tang et al. (2024); Yao & Li (2025); Grislain (2025)). The CCA model generates outputs and attributes credit to items in the deployment-time dataset. This is a good starting point for studying CCA generative models: it is simple to formalize and particularizes credit for individual generated outputs. A drawback is that it only attributes credit to deployment-time data, not training data.

*End-to-end CCA:* To attribute individual generated outputs to particular training data items, one might try applying CCA to the process which first trains a model then interactively answers queries. While promising, analogous versions of DP have proved quite subtle (Dwork & Feldman, 2018; Shariff & Sheffet, 2018; Vietri et al., 2020; Bassily et al., 2018; Kaplan et al., 2023; Naor et al., 2023). CCA versions of these results would only be more difficult, a fact compounded by our result that CCA does not always compose (Theorem 4.2). We leave this direction for future work.

We generalize the definition of CCA to allow a credit-attributing algorithm (the generative model) to take a prompt as input, in addition to the dataset being credited.

**Definition 3.3** (CCA with prompts). *Let $\varepsilon, \delta \geq 0$. An algorithm $\widetilde{A} : 2^{\mathcal{S}} \times \mathcal{X}^* \to \mathcal{Y} \times 2^{\mathcal{S}}$ is $(\varepsilon, \delta)$-CCA if for all prompts $\mathbf{x} \in \mathcal{X}^*$, the algorithm $\widetilde{A}(\cdot, \mathbf{x}) : 2^{\mathcal{S}} \to \mathcal{Y} \times 2^{\mathcal{S}}$ is $(\varepsilon, \delta)$-CCA.*

That is, $\widetilde{A}$ is $(\varepsilon, \delta)$-CCA if for all $\mathbf{x} \in \mathcal{X}^*$, all $S \subseteq \mathcal{S}$, and all $s_i \in S$ the following holds: either $\Pr[\widetilde{A}(S, \mathbf{x}) \text{ credits } s_i] = 1$ or

$$\widetilde{A}(S^{-i}, \mathbf{x}) \approx_{\varepsilon, \delta} \widetilde{A}(S_{-i}, \mathbf{x}),$$

where $\widetilde{A}(S^{-i}, \mathbf{x})$ and $\widetilde{A}(S_{-i}, \mathbf{x})$ are defined as in Definition 3.2.

# 4. Imposing CCA on Next-Token Predictor

A natural starting point for building CCA generative models is to reduce the problem to CCA next-token prediction, following the usual approach for DP LLMs and LLMs generally. First, design a *crediting* next-token predictor $\widetilde{M}$. Then sample outputs from the *crediting rollout* $G^{\widetilde{M}}$ of $\widetilde{M}$

(Definition 4.1). In words, $G^{\widetilde{M}}$ runs $\widetilde{M}$ autoregressively to generate a sequence of tokens $\{y_j\}$ and credit sets for those tokens $\{C_j\}$. Then it returns the concatenated tokens $y_1 \| y_2 \| \dots$ and the union of the credit sets $C_1 \cup C_2 \cup \dots$.

For this to work, we need ($\widetilde{M}$ is CCA) $\implies$ ($G^{\widetilde{M}}$ is CCA). Like differential privacy, CCA would have to *compose* under autoregressive generation.

Unfortunately and perhaps surprisingly, Theorem 4.2 shows that CCA does not compose autoregressively. We give a simple counterexample: for every $\varepsilon \geq 0$ and $\delta < 1$, we describe a $(0, 0)$-CCA next-token predictor $\widetilde{M}$ whose rollout $G^{\widetilde{M}}$ is not $(\varepsilon, \delta)$-CCA.

The counterexample is an instance of Theorem 4.3 that lower bounds the possible CCA parameter $\varepsilon'$ of the rollout of an $\varepsilon$-CCA model $\widetilde{M}$. All else equal, the lower bound on $\varepsilon'$ gets stronger as $\varepsilon \to 0$. This is counterintuitive: one would expect that as $\widetilde{M}$ gets better at attributing credit, the rollout would too. Hence, perhaps taking $\widetilde{M}$ to be $(0, 0)$-CCA as in our counterexample may be the hardest case for autoregressive composition.

Formalizing the preceding discussion, we begin by formally defining the (credit-attributing) rollout $G^{\widetilde{M}}$ of a (credit-attributing) next-token predictor $\widetilde{M}$.

**Definition 4.1** (Credit-attributing rollout). *Let $\widetilde{M} : 2^{\mathcal{S}} \times \mathcal{X}^* \to \mathcal{X} \times 2^{\mathcal{S}}$ be a credit-attributing next-token predictor. The credit-attributing rollout $G^{\widetilde{M}} : 2^{\mathcal{S}} \times \mathcal{X}^* \to \mathcal{X}^* \times 2^{\mathcal{S}}$ of $\widetilde{M}$ is described by Algorithm 2.*

---

**Algorithm 2** $G^{\widetilde{M}}$: crediting rollout of $\widetilde{M}$

---

**input** dataset $S$, prompt $\mathbf{x}_0$
**output** generated sequence $\mathbf{x}$, credit set $C$
1: $y \leftarrow \lambda$ {init. to empty string}
2: $\mathbf{x} \leftarrow \mathbf{x}_0$
3: $C \leftarrow \emptyset$
4: **while** $y \neq \perp$ **do**
5:    $(y, C') \sim \widetilde{M}(S, \mathbf{x})$
6:    $\mathbf{x} \leftarrow \mathbf{x} \| y$
7:    $C \leftarrow C \cup C'$
8: **end while**
9: **Output** $(\mathbf{x}, C)$

---

## 4.1. Counterexample for CCA composition

The following theorem shows that CCA does not compose autoregressively. Even if the starting next-token predictor satisfies the most stringent parameters, $(0, 0)$-CCA, no CCA guarantee can be made about its rollout.

**Theorem 4.2.** *For all $\varepsilon \geq 0$, $0 \leq \delta < 1$, there exists a credit-attributing next-token predictor $\widetilde{M}$ satisfying $(0, 0)$-*

*CCA whose credit-attributing rollout $G^{\widetilde{M}}$ is not $(\varepsilon, \delta)$-CCA.*

*Proof of Theorem 4.2.* Fix the data universe $\mathcal{S} = \{s_1\}$ and the token set $\mathcal{X} = \{\mathtt{a}, \mathtt{b}\}$. Let $0 < p < e^{-\varepsilon} \cdot (1-\delta)$. Define a credit-attributing next-token predictor $\widetilde{M}$ as follows.

For the empty prompt $\mathbf{x} = \lambda$:
$$\widetilde{M}(\{s_1\}, \lambda) \triangleq \widetilde{M}(\emptyset, \lambda) \triangleq \begin{cases} (\mathtt{a}, \emptyset) & \text{w.p. } p \\ (\mathtt{b}, \emptyset) & \text{w.p. } 1-p \end{cases}.$$

For prompt $\mathbf{x} = \mathtt{a}$:
$$\widetilde{M}(\{s_1\}, \mathtt{a}) \triangleq \begin{cases} (\mathtt{a}, \{s_1\}) & \text{w.p. } \frac{1}{2} \\ (\mathtt{b}, \emptyset) & \text{w.p. } \frac{1}{2} \end{cases},$$
$$\widetilde{M}(\emptyset, \mathtt{a}) \triangleq (\mathtt{b}, \emptyset)$$

For prompt $\mathbf{x} = \mathtt{b}$:
$$\widetilde{M}(\{s_1\}, \mathtt{b}) \triangleq (\mathtt{a}, \{s_1\}),$$
$$\widetilde{M}(\emptyset, \mathtt{b}) \triangleq (\mathtt{a}, \emptyset)$$

For all other prompts $\mathbf{x}$:
$$\widetilde{M}(\{s_1\}, \mathbf{x}) \triangleq \widetilde{M}(\emptyset, x) \triangleq (\bot, \emptyset).$$

**The next-token predictor $\widetilde{M}$ is CCA** We now prove that $\widetilde{M}$ is $(0,0)$-CCA by showing that for all prompts $\mathbf{x}$, the restriction $\widetilde{M}(\cdot, \mathbf{x})$ is $(0,0)$-CCA. Let $S = \{s_1\}$ and $S_{-1} = \emptyset$.

For $\mathbf{x} \notin \{\mathtt{a}, \mathtt{b}\}$, $\widetilde{M}(S, \lambda)$ never credits anything:
$$\widetilde{M}(S^{-1}, \mathbf{x}) = \widetilde{M}(S, \mathbf{x}) = \widetilde{M}(\emptyset, \mathbf{x}) = \widetilde{M}(S_{-1}, \mathbf{x}).$$

For $\mathbf{x} = \mathtt{a}$, conditioned on not crediting $s_1$, $\widetilde{M}(S, \mathtt{a})$ always outputs $(\mathtt{b}, \emptyset)$:
$$\widetilde{M}(S^{-1}, \mathtt{a}) = (\mathtt{b}, \emptyset) = \widetilde{M}(\emptyset, \mathtt{a}) = \widetilde{M}(S_{-1}, \mathtt{a}).$$

For $\mathbf{x} = \mathtt{b}$, $\widetilde{M}(S, \mathtt{b})$ always credits $s_1$:
$$\Pr[\widetilde{M}(S, \mathtt{b}) \text{ credits } s_1] = 1.$$

**The crediting rollout $G^{\widetilde{M}}$ is not CCA** We now prove that the credit-attributing rollout $G^{\widetilde{M}}$ of $\widetilde{M}$ is not $(\varepsilon, \delta)$-CCA. It is easy to check that on the empty prompt $\lambda$, $G^{\widetilde{M}}$ is:
$$G^{\widetilde{M}}(\{s_1\}, \lambda) = \begin{cases} (\mathtt{aa}, \{s_1\}) & \text{w.p. } \frac{1}{2}p \\ (\mathtt{ab}, \emptyset) & \text{w.p. } \frac{1}{2}p \\ (\mathtt{ba}, \{s_1\}) & \text{w.p. } 1-p \end{cases},$$
$$G^{\widetilde{M}}(\emptyset, \lambda) = \begin{cases} (\mathtt{ab}, \emptyset) & \text{w.p. } p \\ (\mathtt{ba}, \emptyset) & \text{w.p. } 1-p \end{cases}.$$

Let $S = \{s_1\}$. Conditioning on not crediting $s_1$, we have $\Pr[G^{\widetilde{M}}(S^{-1}, \lambda) = (\mathtt{ab}, \emptyset)] = 1$. In the counterfactual, $\Pr[G^{\widetilde{M}}(S_{-1}, \lambda) = (\mathtt{ab}, \emptyset)] = p < e^{-\varepsilon}(1-\delta)$. Hence, $G^{\widetilde{M}}(S^{-1}, \lambda) \not\approx_{\varepsilon,\delta} G^{\widetilde{M}}(S_{-1}, \lambda)$. $\square$

### 4.2. Lower bound for $\varepsilon$-CCA autoregressive composition

The example in the previous section is a special case of the Theorem 4.3 below, which gives a lower bound of the CCA parameter $\varepsilon'$ that the rollout $G^{\widetilde{M}}$ of a crediting next-token predictor $\widetilde{M}$ can achieve (for $\delta = 0$).

We highlight the counterintuitive dependence on $\varepsilon$: the lower bound on $\varepsilon'$ gets stronger as $\varepsilon \to 0$. However, we caution that $\varepsilon$ is endogenous to the model $\widetilde{M}$. Changing $\varepsilon$ may change other terms in the lower bound. This complicates the interpretation of the already complex theorem statement.

**Theorem 4.3.** *Suppose the next-token predictor $\widetilde{M}$ is $\varepsilon$-CCA and its rollout $G^{\widetilde{M}}$ is $\varepsilon'$-CCA. Given any dataset $S$, document $s_i \in S$ and prompt $\mathbf{x}_0$, either: (1) $\widetilde{M}$ credits $s_i$ with probability 1, or (2) $\varepsilon'$ is at least $\max_{(\mathbf{x}^{-i}, C^{-i})} \underline{\varepsilon}(\mathbf{x}^{-i}, C^{-i})$ where*
$$\underline{\varepsilon}(\mathbf{x}, C) = \left\{ \ln\left( \frac{\prod_{j=1}^{|\mathbf{x}|} \Pr[E_j \mid x_1 \ldots x_{j-1}]}{\Pr[s_i \notin C]} \right) - |\mathbf{x}| \cdot \varepsilon \right\}$$

*and the $\max$ is over $(\mathbf{x}^{-i}, C^{-i}) \in \text{supp}(G^{\widetilde{M}}(S^{-i}, \mathbf{x}))$ (i.e., all outputs where $s_i$ is not credited), the probabilities are over the randomness of $G^{\widetilde{M}}$, and $E_j$ is the event that $s_i$ is not credited in the $j$th step of the rollout (i.e., $s_i \notin C'_j$ for $(x_j, C'_j) \sim \widetilde{M}(S, x_1^{-i} \| \ldots \| x_{j-1}^{-i}))$.*

In the example of the proof of Thm 4.2, we have $\Pr[s_1 \in C] = \frac{1}{2}p$, and for $(\mathbf{x}^{-1}, C^{-1}) = (\mathtt{ab}, \emptyset)$:
$$\prod_{j=1}^{|\mathbf{x}^{-1}|} \Pr[E_j \mid x_1^{-1}, \ldots, x_{j-1}^{-1}]$$
$$= \Pr\left[\widetilde{M}(S, \lambda) \text{ not credit } s_1\right] \Pr\left[\widetilde{M}(S, \mathtt{a}) \text{ not credit } s_1\right]$$
$$= 1 \cdot \frac{1}{2}.$$

Plugging in $\varepsilon = 0$, we get $\varepsilon' \geq \ln\left(\frac{1/2}{p/2}\right) - 2 \cdot 0 = -\ln p$.

## 5. Retrofitting Credit on Existing Autoregressive Model

In this section, we consider the problem of retrofitting credit on an existing autoregressive model: add credit sets as an additional output of the model in a way that satisfies CCA, while not otherwise changing the model and minimizing the *cost* of attributing credit. Starting with a (non-crediting) next-token predictor, a solution to the problem is called a *credit-optimal CCA augmentation*.

This section shows retrofitting is hard: one cannot efficiently implement a credit-optimal CCA augmentation using black-box access to the next-token predictor. It takes

exponentially-many queries in the worst case. In fact, one cannot even approximate the optimal probability of crediting a single document $s \in S$ within a factor of $\pm 1/2$.

This section is organized as follows. §5.1 defines credit-optimal CCA augmentations and the CCA retrofitting problem. §5.2 states our main theorem and gives the high-level overview of the proof. §5.3 gives our construction of a hard-to-retrofit family of models and characterizes the optimal CCA augmentation of those models. §5.4 proves the theorem. Proofs of supporting lemmas are deferred to Appendix B.

### 5.1. Defining the CCA retrofitting problem

This section defines the CCA retrofitting problem (Definition 5.4). Roughly, given a non-crediting model $G^M$ and turn it into a crediting model $\widetilde{G}$ that satisfies CCA while (1) not otherwise changing its behavior (Definition 5.1), and (2) not attributing credit too often (Definition 5.2). We also consider an approximate version of the problem (Definition 5.3).

**Model augmentation**    To capture the first condition, we introduce the concept of model *augmentation*: $\widetilde{G}$ augments $G$ if it preserves the distribution over generated output.

**Definition 5.1.** *Let $G : 2^{\mathcal{S}} \times \mathcal{X}^* \to \mathcal{Y}$ be a model. Let $\widetilde{G} : 2^{\mathcal{S}} \times \mathcal{X}^* \to \mathcal{Y} \times 2^{\mathcal{S}}$ be a crediting model, and $\widetilde{G}_Y : 2^{\mathcal{S}} \times \mathcal{X}^* \to \mathcal{Y}$ be the model obtained by marginalizing $\widetilde{G}$ to its $\mathcal{Y}$ component.*

*$\widetilde{G}$ augments $G$ if $\widetilde{G}_Y = G$. $\widetilde{G}$ is an $(\varepsilon, \delta)$-CCA augmentation of $G$ if also satisfies $(\varepsilon, \delta)$-CCA.*

**Credit optimality**    CCA is trivially satisfied if one always credits everything, taking $C = S$. This would also be worthless: one might as well skip credit attribution altogether. Instead, we want models to be parsimonious in giving credit. This could mean minimizing the expected number of items credited, the probability of crediting a particular data item $s_i$, or a cost function $f$.

We define *credit-optimal* CCA augmentation, the CCA augmentation that minimizes the expected crediting cost $\mathbb{E}[f(C)]$ for a given nonempty dataset $S^*$.[7]

**Definition 5.2.** *A crediting model $\widetilde{G}^*$ is a* credit-optimal *$(\varepsilon, \delta)$-CCA augmentation of $G$ for cost function $f$ and nonempty dataset $S^* \subseteq \mathcal{S}$ if the following hold:*

- *$\widetilde{G}^*$ is an $(\varepsilon, \delta)$-CCA augmentation of $G$*

- *For all prompts $\mathbf{x} \in \mathcal{X}^*$, and for all $(\varepsilon, \delta)$-CCA aug-*

---

*mentations $\widetilde{G}$ of $G$,*

$$\mathbb{E}_{(y,C) \sim \widetilde{G}^*(S^*, \mathbf{x})} \big[ f(C) \big] \leq \mathbb{E}_{(y,C) \sim \widetilde{G}(S^*, \mathbf{x})} \big[ f(C) \big].$$

*When clear from context, we say $\widetilde{G}^*$ is* credit-optimal *or* optimal *for short.*

Looking ahead to our lower bound (Theorem 5.5), we will take $\mathcal{S} \triangleq \{s_1\}$. In this setting, there is only one nonempty dataset ($S^* = \{s_1\}$), and minimizing $\mathbb{E}[f(C)]$ for any nonzero cost function ($f(s_1) > 0$) is equivalent to minimizing $\Pr[s_1 \text{ is credited}]$.

**Approximating models**    Our main result in this section is that it is hard to produce an optimal CCA augmentation of a given model. In fact, we show that it is even hard to *approximate* an optimal CCA augmentation. To make this formal, we now define model approximation.

**Definition 5.3.** *Crediting model $\widetilde{G}'$ is an (additive) $\alpha$-approximation of $\widetilde{G}$ if for all $S$, $\mathbf{x}$, and $s_i \in S$:*

$$\left| \Pr \left[ \widetilde{G}'(S, \mathbf{x}) \text{ credits } s_i \right] - \Pr \left[ \widetilde{G}(S, \mathbf{x}) \text{ credits } s_i \right] \right| \leq \alpha.$$

We emphasize that an approximation $\widetilde{G}^\alpha$ to an optimal CCA augmentation of some model $G$ need not itself be optimal, CCA, nor even an augmentation of $G$.

**CCA-RETROFIT**    We now formally define the CCA retrofitting problem.

**Definition 5.4** (CCA-RETROFIT)**.** *Let $\varepsilon, \delta \geq 0$, $f$ be a cost function, and $S^* \subseteq \mathcal{S}$ be a dataset. Let $\mathcal{M}$ be a collection of next-token predictors.*

*The (exact)* CCA-RETROFIT *problem with respect to $(\varepsilon, \delta, f, S^*, \mathcal{M})$ is as follows: Given oracle access to $M \in \mathcal{M}$, implement an oracle to a credit-optimal $(\varepsilon, \delta)$-CCA augmentation $\widetilde{G}^*$ of $G^M$ (optimality w.r.t. $f$, $S^*$).*

*For $\alpha \geq 0$, the $\alpha$-approximate* CCA-RETROFIT *problem is the following relaxation: Given oracle access to $M$, implement an oracle to an $\alpha$-approximation $\widetilde{G}^\alpha$ of $\widetilde{G}^*$ defined as above.*

We give two clarifying remarks for the above definition. First, oracle access to $M$ provides two types of queries (§2): sampling a token $y \sim M(S, \mathbf{x})$, or evaluating the probability of a token $M(y \mid S, \mathbf{x})$. Observe, that sample access alone is sufficient to sample from the rollout $G^M$.

Second, to *implement an oracle to $\widetilde{G}^*$ (resp. $\widetilde{G}^\alpha$)*, an algorithm is only required to sample from $\widetilde{G}^*(S, \mathbf{x})$ (resp. $\widetilde{G}^\alpha(S, \mathbf{x})$) on any input query $(S, \mathbf{x})$. The algorithm may use $M$ as an oracle in this sampling procedure, and need not produce any explicit representation of the model $\widetilde{G}^*$ (resp. $\widetilde{G}^\alpha$).

## 5.2. Approximate retrofitting requires exponentially-many queries

We now state the main theorem of this section: the number of oracle queries required to solve approximate CCA-RETROFIT, is exponential in a model's output length in the worst case.

**Theorem 5.5** (Hardness of CCA-RETROFIT). *Let $\varepsilon \geq 0$, $\delta = 0$, and $\alpha < 1/2$. Let the data universe be a singleton $\mathcal{S} = \{s_1\}$, and $f$ be any nonzero cost function ($f(s_1) > 0$).*

*There exists a family of models $\{\mathcal{M}_\ell\}_{\ell \geq 2}$ for which:*

*(i) Any algorithm that solves the $\alpha$-approximate CCA-RETROFIT problem for $\mathcal{M}_\ell$ requires $\Omega(2^\ell/\ell \log \ell)$ oracle queries, and*

*(ii) On some input prompt $\mathbf{x}_0$, the rollout of every model $M \in \mathcal{M}_\ell$ always generates outputs of length $\ell + 1$ excluding the $\bot$ at the last position.*

The construction is in Sec. 5.3 and the proof is in Sec. 5.4

The high level approach of the proof is as follows. For a given $\ell$, we construct a family $\mathcal{M}_\ell = \{M_\mathbf{z}\}$ indexed by strings $\mathbf{z} \in \{0,1\}^\ell$. We prove two things about this family:

• *Lemma 5.8 (informal):* Finding $\mathbf{z}$ given oracle access to $M_\mathbf{z}$ requires at least $\Omega(2^\ell)$ queries to $M_\mathbf{z}$.

• *Lemma 5.9 (informal):* Finding $\mathbf{z}$ given oracle access any $\alpha$-approximation of an optimal CCA augmentation of the rollout of $M_\mathbf{z}$ requires only $O(\ell \log \ell)$ queries to the $\alpha$-approximation oracle. The crux of the proof is characterizing the optimal CCA augmentation (Lemma 5.6).

Therefore, any oracle algorithm $A$ that implements an $\alpha$-approximation of an optimal CCA augmentation (i.e., $A$ solves approximate CCA-RETROFIT) must make $\Omega(2^\ell/\ell \log \ell)$ queries to $M_\mathbf{z}$ on average.

**Remark 5.5.1.** *For a relaxation of Definition 5.1 and Definition 5.4, where for all $S \in 2^\mathcal{S}, \mathbf{x} \in \mathcal{X}$, $\mathrm{TV}(\widetilde{G}_Y^*(S, \mathbf{x}), G(S, \mathbf{x})) \leq 2d$, we conjecture that when $d$ vanishes sufficiently fast as a function of $\ell$, a similar lower bound can still be shown to hold by modifying the parameter of the model family we construct in the next section.*

## 5.3. Construction of hard model family

We define a family of next-token predictors $\mathcal{M}_\ell = \mathcal{M}_{\ell,\gamma,\varepsilon} = \{M_\mathbf{z}\}$ with token space $\mathcal{X} = \{0, 1, \bot\}$ and data universe $\mathcal{S} = \{s_1\}$. The family is defined by parameters $\ell \geq 1$, $\gamma \in (0, 1)$, and $\varepsilon \geq 0$, and the models in the family are indexed by strings $\mathbf{z} \in \{0,1\}^\ell$. Looking ahead, $\gamma$ is the probability that $s_1$ is credited in optimal CCA augmentations (Lemma 5.6).

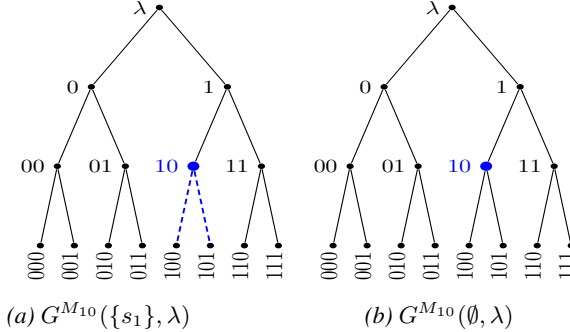

*Figure 1.* Example pair of generation trees with $\ell = 2$ and $\mathbf{z} = 10$. All edges have probability $1/2$ except the dashed blue edges which have probability $1/2 \pm (1 - e^{-\varepsilon}(1 - \gamma))/2$.

The next token distribution $y \sim M_\mathbf{z}$ is defined as follows:

$$M_\mathbf{z}(S, \mathbf{x}) \triangleq \begin{cases} \bot & \text{if } \mathbf{x} \notin \{0,1\}^{\leq \ell} \\ \mathrm{Bern}\left(\frac{1}{2} + \frac{(1-e^{-\varepsilon}(1-\gamma))}{2}\right) & \text{if } S \neq \emptyset \wedge \mathbf{x} = \mathbf{z} \\ \mathrm{Bern}\left(\frac{1}{2}\right) & \text{otherwise} \end{cases}$$

For $\varepsilon \geq 0$ and $\gamma \in (0, 1)$, we have $1 - e^{-\varepsilon}(1 - \gamma) \in (0, 1)$.

We denote by $G_\mathbf{z} \triangleq G^{M_\mathbf{z}}$ the rollout of $M_\mathbf{z}$. $G_\mathbf{z}$ always generates $(\ell + 1)$ bits including the end of generation token $\bot$. On input $\mathbf{x} \neq \mathbf{z}$, the next bit is uniform. On the special input $\mathbf{x} = \mathbf{z}$, the next (and final) bit is biased towards 1 if $S = \mathcal{S} = \{s_1\}$ and uniform if $S = \emptyset$.

Figure 1 shows an example pair of generation trees with $\ell = 2$, where the blue dashed edges are biased, while all other edges are uniform.

The following lemma characterizes the optimal CCA augmentation $\widetilde{G}_\mathbf{z}^*$ of the rollouts $G_\mathbf{z}$ for next-token predictors in the family. If the prompt $\mathbf{x}$ is a prefix of $\mathbf{z}$, denoted $\mathbf{x} \sqsubseteq \mathbf{z}$, then $\widetilde{G}_\mathbf{z}^*$ credits $s_1 \in S$ with constant probability $\gamma$. Otherwise, it does not credit $s_1$.

**Lemma 5.6** (Characterizing the optimal $\widetilde{G}_\mathbf{z}^*$ for the family). *Let $S^* = \{s_1\} = \mathcal{S}$ and let $f$ be any nonzero cost function. For all $M_\mathbf{z} \in \mathcal{M}_{\ell,\gamma,\varepsilon}$, all $\widetilde{G}_\mathbf{z}^*$ optimal $\varepsilon$-CCA augmentations of $G_\mathbf{z}$, and all prompts $\mathbf{x}$:*

$$\Pr\left[\widetilde{G}_\mathbf{z}^*(S^*, \mathbf{x}) \text{ credits } s_1\right] = \begin{cases} \gamma & \mathbf{x} \sqsubseteq \mathbf{z} \\ 0 & otherwise \end{cases}.$$

The proof is deferred to Appendix B.1. To obtain this characterization, we show that any optimal $\widetilde{G}_\mathbf{z}^*$ can be represented as a solution to a linear program and then characterize the optimal solutions of the LP.

**Remark 5.6.1** (Crediting documents despite vanishing impact). *Lemma 5.6 makes us question whether CCA an appropriate definition for credit attribution. To see why, notice*

*that data has a vanishing impact on the models in our family. Concretely,*

$$\mathsf{TV}\big(M_{\mathbf{z}}(\{s_1\}, \mathbf{x}),\ M_{\mathbf{z}}(\emptyset, \mathbf{x})\big) \leq 2^{-\ell}$$

*for every prompt* $\mathbf{x}$*. Functionally, the document* $s_1$ *makes no difference: the models are indistinguishable.*

*Even so, Lemma 5.6 tells us that any optimal $\varepsilon$-CCA augmentation of the rollout $G_{\mathbf{z}}$ is required to credit $s_1$ with constant probability $\gamma > 0$.*

*This violates of our intuition how a credit-attributing model should behave. Perhaps this means that CCA is not a good definition for the problem. Or perhaps the issue goes away if we relax the problem by considering $\delta > 0$ or non-augmentations.*

### 5.4. Proving Theorem 5.5

In this section, we formalize the high-level proof approach discussed in §5.2, and put it all together to prove Theorem 5.5.

First, we formalize the problem of finding $\mathbf{z}$ given access to an oracle computing some randomized function $F_{\mathbf{z}}$.

**Definition 5.7** (FindZ$_{\ell, \gamma, \varepsilon}(F_{\mathbf{z}})$)**.** *Let $F_{\mathbf{z}}$ be an (randomized) function parameterized by $\mathbf{z} \in \{0, 1\}^{\ell}$. An algorithm solves* FindZ$_{\ell, \gamma, \varepsilon}(F_{\mathbf{z}})$ *if it implements the following:*

*Given $(\ell, \gamma, \varepsilon)$ and oracle access to $F_{\mathbf{z}}$, output $\mathbf{z}$ with probability at least $2/3$.*

The complexity of solving FindZ$_{\ell, \gamma, \varepsilon}(F_{\mathbf{z}})$ depends on the function $F_{\mathbf{z}}$ computed by the oracle. The following two lemmas show that exponential queries are required for $F_{\mathbf{z}} = M_{\mathbf{z}}$, but only polynomial queries are required for $F_{\mathbf{z}} = \widetilde{G}_{\mathbf{z}}^{\alpha}$ (any $\alpha$-approximation to an optimal CCA augmentation of the rollout of $M_{\mathbf{z}}$).

**Lemma 5.8.** *Any algorithm solving* FindZ$_{\ell, \gamma, \varepsilon}(M_{\mathbf{z}})$ *requires $\Omega(2^{\ell})$ oracle queries in the worst case over $\mathbf{z} \in \{0, 1\}^{\ell}$.*

**Lemma 5.9.** *For all $\alpha < \gamma/2$, all optimal $\varepsilon$-CCA augmentations $\widetilde{G}_{\mathbf{z}}^*$ of $G_z$, and all $\alpha$-approximations $\widetilde{G}_{\mathbf{z}}^{\alpha}$ of $\widetilde{G}_{\mathbf{z}}^*$:* FindZ$_{\ell, \gamma, \varepsilon}(\widetilde{G}_{\mathbf{z}}^{\alpha})$ *can be solved by an algorithm making $O(\ell \log(12\ell)/(\gamma - 2\alpha)^2)$ oracle queries.*

Together, these lemmas let us prove the theorem. The key observation is that combining an algorithm $A$ solving CCA-RETROFIT with an algorithm $B$ solving FindZ$_{\ell, \gamma, \varepsilon}(\widetilde{G}_{\mathbf{z}}^{\alpha})$ yields an algorithm $B^A$ solving FindZ$_{\ell, \gamma, \varepsilon}(M_{\mathbf{z}})$.

*Proof of Theorem 5.5.* Let $B$ be the oracle algorithm solving FindZ$_{\ell, \gamma, \varepsilon}(\widetilde{G}_{\mathbf{z}}^{\alpha})$ guaranteed by Lemma 5.9 that makes

$$N_B = O(\ell \log(12\ell)/(\gamma - 2\alpha)^2)$$

queries to its oracle. Then $B^A$ is an oracle algorithm that solves FindZ$_{\ell, \gamma, \varepsilon}(M_{\mathbf{z}})$ by making $N_B$ many queries to $A$. By Lemma 5.8, $B^A$ makes $\Omega(2^{\ell})$ queries to its oracle $M_{\mathbf{z}}$ for the worst-case $\mathbf{z}$. Therefore, for the worst-case $\mathbf{z}$, $A$ must make

$$\Omega(2^{\ell} \cdot (\gamma - 2\alpha)^2 / \ell \log(12\ell))$$

queries to its oracle $M_{\mathbf{z}}$.

For any $\mathbf{z}$, by definition, $M_{\mathbf{z}}$ only generates $\perp$ when the input prompt $\mathbf{x}$ has length greater than $\ell + 1$. Therefore, by definition of rollout, $G^{M_{\mathbf{z}}}(S, \mathbf{x}_0)$ with $\mathbf{x}_0 = \lambda$ always generate output of length $\ell + 1$. $\qquad\square$

## 6. Discussion and Open Questions

**Relaxing of CCA retrofitting**  §5 shows that optimal $\varepsilon$-CCA augmentation has undesirable properties: In addition to the computational infeasibility, as Remark 5.6.1 shows, it may also require crediting documents whose impact on the model is vanishingly small. Such requirement violates our intuitions about how credit attribution should behave.

*Question:* Does relaxing CCA-RETROFIT—for example, allowing models that don't exactly preserve the original generation output distribution—get around these undesirable properties?

Our infeasibility result only holds for $\delta = 0$, essentially requiring the CCA counterpart of pure DP. As with DP, allowing $\delta > 0$ (or even other relaxations (Dwork & Rothblum, 2016; Bun & Steinke, 2016; Mironov, 2017; Dong et al., 2022)) might enable qualitatively different results.

*Question:* Does taking $\delta > 0$ (or the CCA counterpart of other DP relaxations) get around the undesirable properties?

**Non-black box retrofitting**  Theorem 5.5 tells us that solving CCA-RETROFIT with *black-box* access to the base model $M$ is hard in the worst case. However, it may still be possible to solve a *non-black box* variant of the problem. Indeed, we have already shown that the non-black box variant is efficient for the family of models $\mathcal{M}_{\ell}$ that we use to prove the hardness result: Lemma 5.6 fully describes the optimal $\varepsilon$-CCA augmentation.

*Question:* Are there efficient algorithms for non-black box retrofitting?

**Credit-optimal CCA augmentations**  Lemma 5.6 characterizes the optimal $\varepsilon$-CCA augmentation for a concrete family of models. We know little about credit-optimal augmentations for general models. Moreover, we required credit-optimality only for a particular cost function $f$ and dataset $S$. A stronger requirement would be optimality for all datasets $S$ or all cost functions $f$ within a given class.

It is not clear that such optima even exist, or whether one would have to settle for Pareto optimality.

*Question:* Which optimality criteria are appropriate and/or easy to optimize over CCA-augmentations in general?

**CCA composition** Theorem 4.2 shows that CCA does not always compose autoregressively: for $G^{\widetilde{M}}$ to be CCA, it does not suffice that $M$ be CCA. That doesn't mean focusing on the next-token predictor $M$ is a dead end.

*Question:* Are there simple conditions on credit-attributing next-token predictor $\widetilde{M}$ that suffices to guarantee CCA of its rollout $G^{\widetilde{M}}$?

## Impact Statement

This paper presents the work that conducts theoretical analysis of the limitation of data attribution of generative AI model. As long-shot speculations, it is possible that our work could influence the public discourse of intellectual property and AI and/or be cited by relevant stakeholders in legal context such as copyright legislation and litigations.

## Acknowledgment

We are grateful for Jason D. Hartline and Ari Holtzman for helpful discussions. We thank the ICML 2026 reviewers for their comments. This research project is supported by NSF-funded Institute for Data, Econometrics, Algorithms and Learning (IDEAL) through the NSF grants ECCS-2216970 and ECCS-2216912.

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

# A. Deferred proofs from Section 4

## A.1. Proof of Theorem 4.3

We restate the theorem for ease of reference.

**Theorem 4.3.** *Suppose the next-token predictor $\widetilde{M}$ is $\varepsilon$-CCA and its rollout $G^{\widetilde{M}}$ is $\varepsilon'$-CCA. Given any dataset $S$, document $s_i \in S$ and prompt $\mathbf{x}_0$, either: (1) $\widetilde{M}$ credits $s_i$ with probability 1, or (2) $\varepsilon'$ is at least $\max_{(\mathbf{x}^{-i}, C^{-i})} \underline{\varepsilon}(\mathbf{x}^{-i}, C^{-i})$ where*

$$\underline{\varepsilon}(\mathbf{x}, C) = \left\{ \ln \left( \frac{\prod_{j=1}^{|\mathbf{x}|} \Pr\left[ E_j \mid x_1 \dots x_{j-1} \right]}{\Pr\left[ s_i \notin C \right]} \right) - |\mathbf{x}| \cdot \varepsilon \right\}$$

*and the $\max$ is over $(\mathbf{x}^{-i}, C^{-i}) \in \mathrm{supp}(G^{\widetilde{M}}(S^{-i}, \mathbf{x}))$ (i.e., all outputs where $s_i$ is not credited), the probabilities are over the randomness of $G^{\widetilde{M}}$, and $E_j$ is the event that $s_i$ is not credited in the $j$th step of the rollout (i.e., $s_i \notin C'_j$ for $(x_j, C'_j) \sim \widetilde{M}(S, x_1^{-i} \| \dots \| x_{j-1}^{-i}))$.*

*Proof.* Fix dataset $S$, document $s_i \in S$, and initial prompt $\mathbf{x}_0$. We introduce the following notations:

- $\mathbf{w} = ((x_1, C'_1), (x_2, C'_2), \dots, (x_{|\mathbf{w}|}, C'_{|\mathbf{w}|}))$ denotes a vector of token-credit set pairs. Its $j$-th component $w_j = (x_j, C'_j)$ corresponds to the output at Line 5 of Algorithm 2 at the $j$-th round: $(x_j, C'_j) \sim \widetilde{M}(S, x_1 \| \dots \| x_{j-1})$. In other words, $\mathbf{w}$ represents a trace of a autoregressive rollout in which $C'_j$ is credited by the next-token predictor $\widetilde{M}$ when the token $x_j$ is generated. We use $s_i \in \mathbf{w}$ to denote the event $s_i \in C'_1 \cup \dots \cup C'_{|\mathbf{w}|}$.

- $P(\cdot)$ is the distribution over traces $\mathbf{w}$ sampled as $\mathbf{w} \sim G^{\widetilde{M}}(S, \mathbf{x}_0)$.

- $Q(\cdot)$ is the distribution over traces $\mathbf{w}$ sampled as $\mathbf{w} \sim G^{\widetilde{M}}(S_{-i}, \mathbf{x}_0)$.

- $P_{\widetilde{M}}(\cdot \mid x_1, \dots, x_{j-1})$ is the distribution over $w_j$ sampled as $w_j \sim \widetilde{M}(S, \mathbf{x}_0 \| x_1 \| \dots \| x_{j-1})$.

- $Q_{\widetilde{M}}(\cdot \mid x_1, \dots, x_{j-1})$ is the distribution over $w_j$ sampled as $w_j \sim \widetilde{M}(S_{-i}, \mathbf{x}_0 \| x_1 \| \dots \| x_{j-1})$.

Let $\mathbf{w}$ be a trace where $s_i$ is never credited: $s_i \notin \mathbf{w}$. By the chain rule,

$$P(\mathbf{w}) = \prod_{j=1}^{|\mathbf{w}|} P(w_j \mid w_1, \dots, w_{j-1}).$$

By definition of the crediting rollout $G^{\widetilde{M}}$ of $\widetilde{M}$ (Algorithm 2), only the tokens $x_j$ sampled at each round are involved in the later calls to $\widetilde{M}$. That is, $w_j$ is conditionally independent of $(C'_1, \dots, C'_{j-1})$ conditioned on $(x_1, \dots, x_{j-1})$. Therefore,

$$P(w_j \mid w_1, \dots, w_{j-1}) = P_{\widetilde{M}}(w_j \mid x_1, \dots, x_{j-1}). \tag{1}$$

Since $s_{-i} \notin \mathbf{w}$, we have $s_{-i} \notin C'_j$. That is, event $E_j$ occurs. Therefore,

$$P_{\widetilde{M}}(w_j \mid x_1, \dots, x_{j-1}) = P_{\widetilde{M}}(w_j \cap E_j \mid x_1, \dots, x_{j-1}) = P_{\widetilde{M}}(w_j \mid E_j, x_1, \dots, x_{j-1}) \cdot P_{\widetilde{M}}(E_j \mid x_1, \dots, x_{j-1}).$$

By definition, the distribution $P_{\widetilde{M}}(\cdot | E_j, x_1, \dots, x_{j-1})$ is the conditional distribution in the CCA definition: $\widetilde{M}(S^{-i}, \mathbf{x} \| x_1 \| \dots \| x_{j-1})$. The corresponding counterfactual distribution is $Q_{\widetilde{M}}(\cdot | x_1, \dots, x_{j-1})$. By assumption that $\widetilde{M}$ satisfies $\varepsilon$-CCA:

$$P_{\widetilde{M}}(w_j \mid E_j, x_1, \dots, x_{j-1}) \geq e^{-\varepsilon} \cdot Q_{\widetilde{M}}(w_j \mid x_1, \dots, x_{j-1}).$$

Combining the previous equations, we get

$$P(\mathbf{w}) \geq e^{-\varepsilon |\mathbf{w}|} \cdot \prod_{j=1}^{|\mathbf{w}|} Q_{\widetilde{M}}(w_j \mid x_1, \dots, x_{j-1}) \cdot \prod_{j=1}^{|\mathbf{w}|} P_{\widetilde{M}}(E_j \mid x_1, \dots, x_{j-1}).$$

By the same logic as (1),

$$\prod_{j=1}^{|\mathbf{w}|} Q_{\widetilde{M}}(w_j \mid x_1, \ldots, x_{j-1}) = \prod_{j=1}^{|\mathbf{w}|} Q(w_j \mid w_1, \ldots, w_{j-1}) = Q(\mathbf{w}).$$

Therefore, for any generation result $(\mathbf{x}^{-i}, C^{-i})$ such that $s_i \notin C^{-i}$, we have

$$P\left((\mathbf{x}^{-i}, C^{-i})\right) = \sum_{\substack{\mathbf{w}=((x_j, C'_j))_j \text{ st.} \\ x_1 \| \cdots \| x_{|\mathbf{w}|} = \mathbf{x}^{-i} \\ \cup_j C'_j = C^{-i}}} P(\mathbf{w}) \geq e^{-\varepsilon |\mathbf{x}^{-i}|} \cdot \sum_{\substack{\mathbf{w}=((x_j, C'_j))_j \text{ st.} \\ x_1 \| \cdots \| x_{|\mathbf{w}|} = \mathbf{x}^{-i} \\ \cup_j C'_j = C^{-i}}} \left[ Q(\mathbf{w}) \cdot \prod_{j=1}^{|\mathbf{w}|} P_{\widetilde{M}}(E_j \mid x_1, \ldots, x_{j-1}) \right]$$

$$= e^{-\varepsilon |\mathbf{x}^{-i}|} \cdot Q\left((\mathbf{x}^{-i}, C^{-i})\right) \cdot \prod_{j=1}^{|\mathbf{w}|} P_{\widetilde{M}}(E_j \mid x_1^{-i}, \ldots, x_{j-1}^{-i}).$$

Since $P\left((\mathbf{x}^{-i}, C^{-i})\right) = P\left((\mathbf{x}^{-i}, C^{-i}) \cap [s_i \notin C]\right)$ by the assumption $s_i \notin C^{-i}$, we conclude that

$$P\left((\mathbf{x}^{-i}, C^{-i}) \mid [s_i \notin C]\right) = \frac{P\left((\mathbf{x}^{-i}, C^{-i})\right)}{P(s_i \notin C)} \geq e^{-\varepsilon |\mathbf{x}^{-i}|} \cdot Q\left((\mathbf{x}^{-i}, C^{-i})\right) \cdot \frac{\prod_{j=1}^{|\mathbf{w}|} P_{\widetilde{M}}(E_j \mid x_1^{-i}, \ldots, x_{j-1}^{-i})}{P(s_i \notin C)}.$$

By assumption $G^{\widetilde{M}}$ is $\varepsilon'$-CCA. By definition of $P$, $Q$, we get

$$e^{\varepsilon'} \geq e^{-\varepsilon |\mathbf{x}^{-i}|} \cdot \frac{\prod_{j=1}^{|\mathbf{w}|} P_{\widetilde{M}}(E_j \mid x_1^{-i}, \ldots, x_{j-1}^{-i})}{P(s_i \notin C)}.$$

Applying a logarithm and taking the maximum over generations $(\mathbf{x}^{-i}, C^{-i})$ such that $s_i \notin C^{-i}$ yields the inequality in the theorem statement. $\square$

## B. Deferred proofs from Section 5

### B.1. Proof of Lemma 5.6

We start from the definition $\varepsilon$-CCA augmentation and reduce the constraints to an essential subsets utilizing the instance-specific properties. We then characterize the solution satisfying these essential constraints.

By Definition 5.2, any optimal $\varepsilon$-CCA augmentation $\widetilde{G}_{\mathbf{z}}^*$ of $G_{\mathbf{z}}$ solves the following optimization problem for all $S'$, all $s_1 \in S'$, and all $\mathbf{x}$,

$$\min_{(y,C) \sim \widetilde{G}_{\mathbf{z}}(S', \mathbf{x})} \Pr \left[ s_1 \in C \right] \tag{2}$$
$$s.t. \text{ (a)} \vee \text{ (b)}$$

where

(a) $P(s_1 \in C) = 1$,

(b) $P(\cdot \mid s_1 \notin C) \approx_\varepsilon Q$,

$P$ is the output distribution of $\widetilde{G}_{\mathbf{z}}(S', \mathbf{x})$, and $Q$ is the output distribution of $\widetilde{G}_{\mathbf{z}}(S' \setminus \{s_1\}, \mathbf{x})$.

**Claim 1.** *Given $\mathcal{S} = \{s_1\}$. If $\widetilde{G}_{\mathbf{z}}^*$ solves the following optimization problem for $S = \{s_1\}$ and all $\mathbf{x}$:*

$$\min_{(y,C) \sim \widetilde{G}_{\mathbf{z}}(S, \mathbf{x})} \Pr \left[ s_1 \in C \right] \tag{3}$$
$$s.t. \text{ (not (a))} \wedge \text{ (b)}.$$

*and for $S' = \emptyset$ gives no credit, i.e., for all $\mathbf{x}$,*

$$\Pr_{(y,C)\sim\widetilde{G}^*_{\mathbf{z}}(\emptyset,\mathbf{x})}\left[s_1 \in C\right] = 0,$$

*then $\widetilde{G}^*_{\mathbf{z}}$ is an optimal $\varepsilon$-CCA augmentation of $G_{\mathbf{z}}$.*

*Proof.* In fact, since $\mathcal{S} = \{s_1\}$, the only possible datasets are $\emptyset$ and $S = \{s^*\} = \mathcal{S}$.

For $S' = \emptyset$, the CCA condition is always satisfied vacuously and $\widetilde{G}^*_{\mathbf{z}}$ is already crediting $s_1$ with minimum probability 0.

For $S' = S = \{s_1\}$, consider any optimal $\varepsilon$-CCA augmentation $\widetilde{G}^{*\prime}_{\mathbf{z}}$ of $G_{\mathbf{z}}$. For all $\mathbf{x}$, $\widetilde{G}^{*\prime}_{\mathbf{z}}(S,\mathbf{x})$ solves

$$\min_{(y,C)\sim\widetilde{G}_{\mathbf{z}}(S,\mathbf{x})} \Pr\left[s_1 \in C\right] \tag{4}$$
$$s.t.\ (a) \vee (b).$$

If (a) holds for some $\mathbf{x}$, then $\Pr_{(y,C)\sim\widetilde{G}^{*\prime}_{\mathbf{z}}(S,\mathbf{x})}\left[s_1 \in C\right] = P(s_1 \in C) = 1$. Hence $\widetilde{G}^{*\prime}_{\mathbf{z}}$ is dominated by any $\widetilde{G}^*_{\mathbf{z}}$ that solves (3) for all $\mathbf{x}$, since 1 is the maximum possible value of the objective being a probability.

Therefore, any $\widetilde{G}^*_{\mathbf{z}}$ that solve (3) for all $\mathbf{x}$ and gives no credit for $S' = \emptyset$ is an optimal $\varepsilon$-CCA augmentation of $G_{\mathbf{z}}$. $\quad\square$

Next we show that for $S = \{s_1\}$, the crediting probability of $s_1$ subject to CCA for each individual $\mathbf{x}$ can be optimized separately using a linear program. Fix any $\mathbf{x}$, the following linear program maximizes the total probability $R$ of not giving credit, by choosing for each $y$ the probability $r_y$ of not giving credit conditional on the generation output being $y$, subject to the constraints of $\varepsilon$-CCA.

**Claim 2.** *For any fixed $\mathbf{x}$, the optimization problem (3) is equivalent to the following linear program with decision variables $R, \{r_y\}_y$,*

$$\max R$$
$$s.t.\ e^{-\varepsilon}q_y \cdot R \leq r_y p_y \leq e^{\varepsilon}q_y \cdot R \quad \forall y \in \mathcal{X}^*$$
$$R = \sum_{y \in \mathcal{X}^*} r_y p_y \tag{5}$$
$$r_y \in [0,1]$$

*where*

$$p_y \triangleq \Pr_{y'\sim G_{\mathbf{z}}(\{s_1\},\mathbf{x})}[y' = y],\ q_y \triangleq \Pr_{y'\sim G_{\mathbf{z}}(\emptyset,\mathbf{x})}[y' = y]$$

*are the probability of outputting $y$ by the original and counterfactual distributions of model $G_{\mathbf{z}}$ being augmented on the prompt $\mathbf{x}$.*

*Proof.* For $S^* = \{s_1\}$ and fix $\mathbf{x}$, let $P, Q$ be the original and counterfactual distributions of $\widetilde{G}_{\mathbf{z}}(S^*, \mathbf{x})$. The component for $\mathbf{x}$ of (3) is equivalent to

$$\min P(s_1 \in C)$$
$$s.t.\ e^{-\varepsilon}Q(y,C) \leq \frac{P((y,C) \wedge s_1 \notin C)}{P(s_1 \notin C)} \leq e^{\varepsilon}Q(y,C) \quad \forall y \in \mathcal{X}^*, C \in \{\{s_1\},\emptyset\}. \tag{6}$$

By definition, $Q$ is the output distribution of $\widetilde{G}_{\mathbf{z}}(S \setminus \{s_1\}, \mathbf{x}) = \widetilde{G}_{\mathbf{z}}(\emptyset, \mathbf{x})$ and hence $Q(y,\{s_1\}) = 0$. At the same time, $P((y,\{s_1\}) \wedge s_1 \notin \{s_1\}) = 0$. Therefore, the constraints in (6) on $C = \{s_1\}$ are automatically satisfied. $Q(y,\{s_1\}) = 0$ further implies that

$$Q(y,\emptyset) = Q(y,\emptyset) + Q(y,\{s_1\}) = \sum_{C\subseteq S} Q(y,C) = Q_Y(y).$$

Also note that $P(y, \emptyset) = P(\emptyset \mid y)P_Y(y)$, and

$$1 - P(s_1 \in C) = P(s_1 \notin C) = \sum_{y \in \mathcal{X}^*} P(y, \emptyset) = \sum_{y \in \mathcal{X}^*} P(\emptyset \mid y)P_Y(y)$$

By the fact that $\widetilde{G}_{\mathbf{z}}^*$ is an augmentation of $G_{\mathbf{z}}$ and Definition 5.1 of the augmentation, $P_Y(y) = p_y$, $Q_Y(y) = q_y$. Therefore, (6) is equivalent to the

$$\min 1 - \sum_{y \in \mathcal{X}^*} P(\emptyset \mid y)p_y$$

$$s.t. \ e^{-\varepsilon}q_y \leq \frac{P(\emptyset \mid y)p_y}{\sum_{y \in \mathcal{X}^*} P(\emptyset \mid y)P_y} \leq e^{\varepsilon}q_y \ \forall y \in \mathcal{X}^*. \tag{7}$$

Let $r_y = P(\emptyset \mid y)$ be the probability of not crediting $s_1$ given the output being $y$, changing the $\min$ to $\max$ by removing constants and flipping the sign on the objective, (7) becomes

$$\max \sum_{y \in \mathcal{X}^*} r_y p_y$$

$$s.t. \ e^{-\varepsilon}q_y \leq \frac{r_y p_y}{\sum_{y \in \mathcal{X}^*} r_y p_y} \leq e^{\varepsilon}q_y \ \forall y \in \mathcal{X}^* \tag{8}$$

$$r_y \in [0, 1]$$

where $\{r_y\}_y$ are the decision variables. Let $R = \sum_{y \in \mathcal{X}^*} r_y p_y$ and rearranging (8), we get the linear program (5) $\qquad \square$

It suffices to show that for any $\mathbf{x}$, the optimal solution $R^*, \{r_y^*\}_y$ to the linear program (5) satisfies

$$1 - \Pr[\widetilde{G}_{\mathbf{z}}^*(\{s_1\}, \mathbf{x}) \text{ credits } s_1] = R^* = \sum_y r_y^* p_y = \begin{cases} 1 - \gamma & \mathbf{x} \sqsubseteq \mathbf{z} \\ 1 & otherwise \end{cases}.$$

**Claim 3.** *For any* $\mathbf{x}$*, an optimal solution* $R^*, \{r_y^*\}_y$ *to the linear program* (5) *satisfies*

$$R^* = \sum_{y \in \mathcal{X}^*} r_y^* p_y = \begin{cases} 1 - \gamma & \mathbf{x} \sqsubseteq \mathbf{z} \\ 1 & otherwise \end{cases}.$$

*Proof.* We first show the "otherwise" case when $\mathbf{x} \not\sqsubseteq \mathbf{z}$. Then the proof for the case $\mathbf{x} \sqsubseteq \mathbf{z}$ is divided into two steps: In Step 1 we show that $1 - \gamma$ is an upper bound of the optimal objective value $R^*$. In Step 2, we show that there exists $\{r_y\}_y$ that attain the objective value $1 - \gamma$.

**"Otherwise" case when $\mathbf{x} \not\sqsubseteq \mathbf{z}$** First note that by definition of $M_{\mathbf{z}}$ and autoregressive composition, $G_{\mathbf{z}}(\{s_1\}, \mathbf{x}) = G_{\mathbf{z}}(\emptyset, \mathbf{x})$, i.e., $P(y) = Q(y)$ for all $y \in \mathcal{X}^*$ and hence $p_y = q_y$. Taking $r_y^* = 1$ for all $y \in \mathcal{X}^*$ is the optimal solution to the linear program (5) that yields objective value $\sum_{y \in \mathcal{X}^*} 1 \cdot p_y = \sum_{y \in \mathcal{X}^*} p_y = 1$.

**Step 1 for $\mathbf{x} \sqsubseteq \mathbf{z}$** Rearranging the constraints we know that

$$\forall y \in \mathcal{X}^*, R \leq r_y e^{\varepsilon} \cdot \frac{p_y}{q_y}.$$

Let $y^* = \arg\min_y p_y/q_y$, we have an upper bound of the optimal objective value

$$R \leq r_{y^*} e^{\varepsilon} \cdot \frac{p_{y^*}}{q_{y^*}} \leq e^{\varepsilon} \cdot \frac{p_{y^*}}{q_{y^*}} \triangleq \overline{R^*}.$$

By definition of $M_{\mathbf{z}}$ and its rollout $G_{\mathbf{z}}$, we have $y^* = \mathbf{z} \parallel 0$, $p_{y^*}/q_{y^*} = e^{-\varepsilon}(1 - \gamma)$ and $\overline{R^*} = 1 - \gamma$.

**Step 2 for $\mathbf{x} \sqsubseteq \mathbf{z}$**   We now claim that there exists $\{r_y\}_y$ feasible for the constraints of (5) such that $R = \sum_{y \in \mathcal{X}^*} r_y p_y = \overline{R}^* = 1 - \gamma$, and the optimal has objective value $1 - \gamma$.

It suffices to show that there exists $\{r_y\}_y$ such that $R = \sum_{y \in \mathcal{X}^*} r_y p_y = \overline{R}^*$, and for all $y \in \mathcal{X}^*$,

$$e^{-\varepsilon} q_y \leq \frac{r_y p_y}{\overline{R}^*} \leq e^{\varepsilon} q_y, \ 0 \leq r_y \leq 1,$$

i.e.,

$$e^{-\varepsilon} \cdot \overline{R}^* \cdot \frac{q_y}{p_y} \leq r_y \leq \min\left\{ e^{\varepsilon} \cdot \overline{R}^* \cdot \frac{q_y}{p_y}, 1 \right\}, \tag{9}$$

since $q_y, p_y, \overline{R}^* > 0$. For $y \in \mathcal{X}^*$, let

$$\underline{r}_y \triangleq e^{-\varepsilon} \overline{R}^* \cdot \frac{q_y}{p_y}, \ \overline{r}_y \triangleq \min\left\{ e^{\varepsilon} \overline{R}^* \cdot \frac{q_y}{p_y}, 1 \right\}$$

be the lower and upper bounds of $r_y$ given by (9) respectively.

To show the existence of such $\{r_y\}_y$, it suffices to show that

$$\sum_{y \in \mathcal{X}^*} \underline{r}_y p_y \leq \overline{R}^* \leq \sum_{y \in \mathcal{X}^*} \overline{r}_y p_y \triangleq \overline{R}. \tag{10}$$

This is because if the above inequalities holds, then one can obtain $\{r_y\}_y$ starting from initializing $r_y = \underline{r}_y$ for all $y \in \mathcal{X}^*$ and then saturate $r_y$ for $y \in \mathcal{X}^*$ until $R = \sum_y r_y p_y$ reaches $\overline{R}^*$. To see this: The left hand side inequality guarantees that by initializing $r_y = \underline{r}_y$ for all $y$, $\sum_y r_y p_y \leq \overline{R}^*$. The right hand side inequality guarantees that when all $r_y$ are increased to its upper limit $\overline{r}_y$, $\overline{R}^* \leq \sum_y r_y p_y$. Therefore, there must exists some intermediate $\{r_y\}_y$ with $\underline{r}_y \leq y \leq \overline{r}_y$ for each $y$ such that $\sum_y r_y p_y = \overline{R}^*$.

Now we show (10) holds. First note that the left hand side inequality of (10) always holds,

$$\sum_{y \in \mathcal{X}^*} \underline{r}_y p_y = \sum_{y \in \mathcal{X}^*} e^{-\varepsilon} \overline{R}^* \cdot \frac{q_y}{p_y} \cdot p_y \leq \sum_{y \in \mathcal{X}^*} e^{-\varepsilon} \overline{R}^* \cdot q_y = e^{-\varepsilon} \overline{R}^* \sum_{y \in \mathcal{X}^*} q_y = e^{-\varepsilon} \overline{R}^* \leq \overline{R}^*.$$

Next we show that the right hand side inequality of (10) holds for the instance. Let $y' = \mathbf{z} \| 1$. By definition of $M_{\mathbf{z}}$ and its rollout $G_{\mathbf{z}}$, we have

- $q_{y'} = (1/2)^{\ell+1}$, $p_{y'} = (1/2)^{\ell+1}(2 - e^{-\varepsilon}(1 - \gamma))$ and $\overline{r}_{y'} = \min\left\{ e^{\varepsilon}(1 - \gamma)/(2 - e^{-\varepsilon}(1 - \gamma)), 1 \right\}$,

- $q_y = p_y = (1/2)^{\ell+1}$ for all $y \in \mathcal{X}^* \setminus \{y', y^*\}$.

Therefore,

$$\begin{aligned}
\overline{R} - \overline{r}_{y'} p_{y'} - \overline{r}_{y^*} p_{y^*} &= \sum_{y \in \mathcal{X}^* \setminus \{y', y^*\}} \overline{r}_y p_y \\
&= \sum_{y \in \mathcal{X}^* \setminus \{y', y^*\}} \min\left\{ e^{\varepsilon} \overline{R}^* \cdot \frac{q_y}{p_y}, 1 \right\} \cdot p_y \\
&= \sum_{y \in \mathcal{X}^* \setminus \{y', y^*\}} \min\left\{ e^{\varepsilon} \overline{R}^*, 1 \right\} p_y.
\end{aligned}$$

Recall that $\overline{R}^* = 1 - \gamma$ for the instance, therefore we have

$$\overline{R} - \overline{r}_{y'} p_{y'} - \overline{r}_{y^*} p_{y^*} = \min\{ e^{\varepsilon}(1 - \gamma), 1 \} \cdot \sum_{y \in \mathcal{X}^* \setminus \{y', y^*\}} p_y = \min\{ e^{\varepsilon}(1 - \gamma), 1 \} \cdot \left( 1 - \frac{1}{2^{\ell}} \right). \tag{11}$$

Note that $r_{y^*} = \min\left\{e^\varepsilon(1-\gamma)/\left(e^{-\varepsilon}\cdot(1-\gamma)\right), 1\right\} = \min\{e^{2\varepsilon}, 1\} = 1$ and $p_{y^*} = (1/2)^{\ell+1}(e^{-\varepsilon}(1-\gamma))$. Therefore we have

$$\overline{R} - \overline{r}_{y'}p_{y'} = \min\{e^\varepsilon(1-\gamma), 1\}\cdot\left(1 - \frac{1}{2^\ell}\right) + \frac{1}{2^{\ell+1}}(e^{-\varepsilon}(1-\gamma)).$$

Recall that $\overline{r}_{y'} = \min\left\{e^\varepsilon(1-\gamma)/(2 - e^{-\varepsilon}(1-\gamma)), 1\right\}$, and we have $2 - e^{-\varepsilon}(1-\gamma) > 1$ as $e^{-\varepsilon} \leq 1$ and $1 - \gamma < 1$.

- Case 1: $e^\varepsilon(1-\gamma) \leq 1$. We have $\overline{r}_{y'} = e^\varepsilon(1-\gamma)/(2 - e^{-\varepsilon}(1-\gamma))$,

$$\overline{R} = e^\varepsilon(1-\gamma)\cdot\left(1 - \frac{1}{2^\ell}\right) + \frac{1}{2^{\ell+1}}(e^{-\varepsilon}(1-\gamma)) + \frac{e^\varepsilon(1-\gamma)}{2 - e^{-\varepsilon}(1-\gamma)}\cdot\frac{1}{2^{\ell+1}}(2 - e^{-\varepsilon}(1-\gamma))$$

$$= e^\varepsilon(1-\gamma)\cdot\left(1 - \frac{1}{2^\ell}\right) + \frac{1}{2^{\ell+1}}(e^{-\varepsilon}(1-\gamma)) + e^\varepsilon(1-\gamma)\cdot\frac{1}{2^{\ell+1}}$$

$$= (1-\gamma)\cdot\left(e^\varepsilon - e^\varepsilon\cdot\frac{1}{2^\ell} + \frac{1}{2^{\ell+1}}\left(e^{-\varepsilon} + e^\varepsilon\right)\right).$$

Note that $e^{-\varepsilon} + e^\varepsilon \geq 2$ for all $\varepsilon > 0$,

$$\overline{R} \geq (1-\gamma)\cdot\left(e^\varepsilon - e^\varepsilon\cdot\frac{1}{2^\ell} + \frac{1}{2^{\ell+1}}\cdot 2\right)$$

$$= (1-\gamma)\cdot\left(e^\varepsilon - (e^\varepsilon - 1)\cdot\frac{1}{2^\ell}\cdot\right)$$

$$= (1-\gamma)\cdot\left(e^\varepsilon - 1 + 1 - (e^\varepsilon - 1)\cdot\frac{1}{2^\ell}\cdot\right)$$

$$= (1-\gamma)\cdot\left((e^\varepsilon - 1)\left(1 - \frac{1}{2^\ell}\right) + 1\right).$$

Since $e^\varepsilon - 1 \geq 0$ for all $\varepsilon > 0$, we have

$$\overline{R} \geq (1-\gamma)\cdot 1 = 1 - \gamma = \overline{R^*}.$$

- Case 2: $1 < e^\varepsilon(1-\gamma) \leq 2 - e^{-\varepsilon}(1-\gamma)$. We have $\overline{r}_{y'} = e^\varepsilon(1-\gamma)/(2 - e^{-\varepsilon}(1-\gamma))$, and

$$\overline{R} = 1\cdot\left(1 - \frac{1}{2^\ell}\right) + \frac{1}{2^{\ell+1}}(e^{-\varepsilon}(1-\gamma)) + \frac{e^\varepsilon(1-\gamma)}{2 - e^{-\varepsilon}(1-\gamma)}\cdot\frac{1}{2^{\ell+1}}(2 - e^{-\varepsilon}(1-\gamma))$$

$$= 1 - \frac{1}{2^\ell} + \frac{1}{2^{\ell+1}}(e^{-\varepsilon}(1-\gamma)) + e^\varepsilon(1-\gamma)\cdot\frac{1}{2^{\ell+1}}$$

$$= 1 - \frac{1}{2^\ell} + \frac{1}{2^{\ell+1}}(e^{-\varepsilon} + e^\varepsilon)\cdot(1-\gamma).$$

Since $e^{-\varepsilon} + e^\varepsilon \geq 2$ for all $\varepsilon > 0$, therefore

$$\overline{R} \geq 1 - \frac{1}{2^\ell} + \frac{1}{2^{\ell+1}}\cdot 2\cdot(1-\gamma) = 1 + \frac{1}{2^\ell}\cdot(-1 + 1 - \gamma) = 1 - \frac{\gamma}{2^\ell} \geq 1 - \gamma = \overline{R^*}.$$

- Case 3: $2 - e^{-\varepsilon}(1-\gamma) < e^\varepsilon(1-\gamma)$. We have $\overline{r}_{y'} = 1$, and

$$\overline{R} = 1\cdot\left(1 - \frac{1}{2^\ell}\right) + \frac{1}{2^{\ell+1}}(e^{-\varepsilon}(1-\gamma)) + \frac{1}{2^{\ell+1}}(2 - e^{-\varepsilon}(1-\gamma))$$

$$= 1 - \frac{1}{2^\ell} + \frac{1}{2^{\ell+1}}\cdot 2$$

$$= 1 \geq 1 - \gamma = \overline{R^*}.$$

Finally, we conclude that the right hand side inequality of (10) holds and hence for $\mathbf{x} \sqsubseteq \mathbf{z}$, an optimal solution $\{r_y^*\}_y$ satisfies

$$R^* = \sum_{y \in \mathcal{X}^*} r_y^* p_y = \overline{R^*} = 1 - \gamma.$$

$\square$

Therefore, for $\mathbf{x} \sqsubseteq \mathbf{z}$, we have $\Pr\left[\widetilde{G}_\mathbf{z}^*(S^*, \mathbf{x}) \text{ credits } s_1\right] = 1 - R^* = \gamma$. This conclude the proof of Lemma 5.6.

$\square$

### B.2. Proof of Lemma 5.8

*Proof.* We show that the lowerbound holds even for algorithms that have access to the full distribution of $M_\mathbf{z}(S, \mathbf{x})$ when querying $M_\mathbf{z}$ on $(S, \mathbf{x})$. Such query is stronger than the queries an oracle algorithm can make as defined in Section 2: One can simulate the two types of queries with the full distribution of $M_\mathbf{z}(S, \mathbf{x})$.

Given (randomized) algorithm $A$ that always makes fewer than $N = 2^\ell/3 - 1 = \Omega(2^\ell)$ queries to $M_\mathbf{z}$. Let $\mathcal{A}$ be the set of deterministic algorithms that always make fewer than $N$ queries to $M_\mathbf{z}$. By Yao's Minimax Lemma (Yao, 1977), the error probabilitity of $A$ on the worst $\mathbf{z}$

$$\Pr_A[\mathbf{z} \neq A(M_\mathbf{z})] = \max_{M \in \mathcal{M}_{\ell,\gamma,\varepsilon}} \mathbb{E}_A[\mathbf{1}[A(M_\mathbf{z}) \neq \mathbf{z}]]$$

$$\geq \min_{A' \in \mathcal{A}} \mathbb{E}_{M_\mathbf{z} \sim \mathcal{D}}[\mathbf{1}[A'(M_\mathbf{z}) \neq \mathbf{z}]]$$

for any distribution $\mathcal{D}$ over $\mathcal{M}_{\ell,\gamma,\varepsilon}$.

By definition the output distribution of $M_\mathbf{z}(S, \mathbf{x})$ is identical for all $S, \mathbf{x} \in \{0,1\}^\ell$ except $S = S^* = \{s_1\}, \mathbf{x} = \mathbf{z}$. Therefore, any deterministic $A' \in \mathcal{A}$, even with access to the full distribution of $M_\mathbf{z}(S, \mathbf{x})$ after querying $M_\mathbf{z}$ on $(S, \mathbf{x})$ can not update its state before making the query $M_\mathbf{z}(\{s_1\}, \mathbf{z})$. Therefore, $A'$ must make queries to $M_\mathbf{z}$ according to a fixed sequence $T$ of prompts with length at most $N$ until it makes the query $M_\mathbf{z}(\{s_1\}, \mathbf{z})$.

Let $\mathcal{D}$ be the uniform distribution over $\mathcal{M}_{\ell,\gamma,\varepsilon}$, we have

$$\mathbb{E}_{M_\mathbf{z} \sim \mathcal{D}}[\mathbf{1}[A(M_\mathbf{z}) \neq \mathbf{z}]] = \sum_{\mathbf{z} \in \{0,1\}^\ell} \frac{1}{2^\ell} \cdot \mathbf{1}[A'(M_\mathbf{z}) \neq \mathbf{z}]$$

$$\geq \sum_{\mathbf{z} \in \{0,1\}^\ell \setminus T} \frac{1}{2^\ell} \cdot \mathbf{1}[A'(M_\mathbf{z}) \neq \mathbf{z}].$$

Let $\mathbf{z}'$ be the output of $A'$ when it reaches the end of the sequence $T$ and has not made the query $M_\mathbf{z}(\{s_1\}, \mathbf{z})$. For $\mathbf{z} \notin T$, we know that $A'(M_\mathbf{z})$ reaches the end of $T$ and outputs $\mathbf{z}'$, therefore

$$\mathbb{E}_{M_\mathbf{z} \sim \mathcal{D}}[\mathbf{1}[A(M_\mathbf{z}) \neq \mathbf{z}]] \geq \sum_{\mathbf{z} \in \{0,1\}^\ell \setminus T} \frac{1}{2^\ell} \cdot \mathbf{1}[A'(M_\mathbf{z}) \neq \mathbf{z}]$$

$$= \frac{1}{2^\ell} \sum_{\mathbf{z} \in \{0,1\}^\ell \setminus T} \mathbf{1}[\mathbf{z}' \neq \mathbf{z}].$$

By assumption that $T$ has length at most $N = 2^\ell/3 - 1$, we have

$$\mathbb{E}_{M_\mathbf{z} \sim \mathcal{D}}[\mathbf{1}[A(M_\mathbf{z}) \neq \mathbf{z}]] \geq \frac{1}{2^\ell} \sum_{\mathbf{z} \in \{0,1\}^\ell \setminus T} \mathbf{1}[\mathbf{z}' \neq \mathbf{z}]$$

$$\geq \frac{1}{2^\ell} \sum_{\mathbf{z} \in \{0,1\}^\ell \setminus T \setminus \{\mathbf{z}'\}} \mathbf{1}[\mathbf{z}' \neq \mathbf{z}]$$

$$\geq \frac{1}{2^\ell} \cdot (2^\ell - N - 1) = \frac{1}{3}$$

Hence $A$ must have error probability $\Pr_A[\mathbf{z} \neq A(M_{\mathbf{z}})] > \frac{1}{3}$. Therefore, any algorithm solving $\mathsf{FindZ}_{\ell,\gamma,\varepsilon}(M_{\mathbf{z}})$ requires $\Omega(2^\ell)$ queries in the worst case.

$\square$

### B.3. Proof of Lemma 5.9

*Proof.* Let $c(\mathbf{x}) \triangleq \Pr\left[\widetilde{G}_{\mathbf{z}}^*(S, \mathbf{x}) = \{s_1\}\right]$. By Lemma 5.6, we know that $c(\mathbf{x}) = \gamma$ if $\mathbf{x}$ is a prefix of $\mathbf{z}$, and $c(\mathbf{x}) = 0$ otherwise. Consider the following algorithm that starts from the empty string $\mathbf{x} = \lambda$ and iteratively recovers $\mathbf{z}$ by appending one bit at a time. To decide which bit to append, the algorithm utilizes the seperation of $c(\mathbf{x} \| x)$ between when $\mathbf{x} \| x$ is a prefix of $\mathbf{z}$ and when it is not.

---

**Algorithm 3** Algorithm solving $\mathsf{FindZ}_{\ell,\gamma,\varepsilon}(\widetilde{G}_{\mathbf{z}}^\alpha)$

---

**input** oracle access to $\widetilde{G}_{\mathbf{z}}^\alpha$
**output** sequence $\mathbf{z}$
  1: $\mathbf{x} \leftarrow \lambda$
  2: $k \leftarrow 1$
  3: **while** $k < \ell$ **do**
  4:      $c_0 \leftarrow$ estimate of $c^\alpha(\mathbf{x} \| 0)$ by querying $\widetilde{G}_{\mathbf{z}}^\alpha$
  5:      $c_1 \leftarrow$ estimate of $c^\alpha(\mathbf{x} \| 1)$ by querying $\widetilde{G}_{\mathbf{z}}^\alpha$
  6:      **if** $c_0 > c_1$ **then**
  7:          $\mathbf{x} \leftarrow \mathbf{x} \| 0$
  8:      **else**
  9:          $\mathbf{x} \leftarrow \mathbf{x} \| 1$
 10:      **end if**
 11:      $k \leftarrow k + 1$
 12: **end while**
 13: **Output**($\mathbf{x}$)

---

For the estimation of $c^\alpha(\mathbf{x} \| 0)$ and $c^\alpha(\mathbf{x} \| 1)$ in Line 4 and 5, choose the parameters of additive error $\tau < (\gamma - 2\alpha)/2$ and success probilitity $1 - \beta = 1 - 1/(6\ell)$. Applying Lemma B.1, the estimations with parameter $\tau$ and $\beta$ can be done with using

$$O(\tau^{-2} \log(2/\beta)) = O\left(\left(\frac{\gamma - 2\alpha}{2}\right)^{-2} \cdot \log\left(\frac{2}{1/6\ell}\right)\right) = O(4/(\gamma - 2\alpha)^2 \cdot \log(12\ell))$$

oracle queries to $\widetilde{G}_{\mathbf{z}}^\alpha$. Then, the total number of queries the algorithms makes is

$$O(2\ell \cdot 4/(\gamma - 2\alpha)^2 \cdot \log(12\ell)) = O(\ell \log(12\ell)/(\gamma - 2\alpha)^2).$$

The loop from Line 3 to Line 12 iterates for $\ell$ times, and there are 2 etimations in each iteration. By union bound, the probability that all estimations succeed is at least $1 - 2\ell\beta = 2/3$, in which case the algorithm correctly outputs $\mathbf{z}$. We claim that when all estimations in Algorithm 3 succeed, the algorithm correctly outputs $\mathbf{z}$. In fact, we show by induction that, when all estimations succeed, the invariant $\mathbf{x} \sqsubseteq \mathbf{z}$ always holds before the algorithm outputs $\mathbf{x}$.

- Base case: $\mathbf{x} = \lambda$. The invariant naturally holds since $\lambda \sqsubseteq \mathbf{z}$.

- Inductive step: Assume the invariant holds for $\mathbf{x}$.

  Let $x \in \{0, 1\}$ such that $\mathbf{x} \| x \sqsubseteq \mathbf{z}$ and $\bar{x}$ be the other bit such that $\mathbf{x} \| \bar{x} \not\sqsubseteq \mathbf{z}$.

  By assumption that the estimations succeed, we have

$$c_x > c^\alpha(\mathbf{x} \| x) - (\gamma - 2\alpha)/2, \; c_{\bar{x}} < c^\alpha(\mathbf{x} \| \bar{x}) + (\gamma - 2\alpha)/2.$$

Further, by assumptions that $\widetilde{G}_{\mathbf{z}}^{\alpha}$ is an $\alpha$-approximation of $\widetilde{G}_{\mathbf{z}}^{*}$, we have

$$c^{\alpha}(\mathbf{x} \parallel x) > c(\mathbf{x} \parallel x) - \alpha, \ c^{\alpha}(\mathbf{x} \parallel \bar{x}) < c(\mathbf{x} \parallel \bar{x}) + \alpha.$$

Combining the above inequalities, we have

$$c_x > c(\mathbf{x} \parallel x) - \alpha - (\gamma - 2\alpha)/2, \ c_{\bar{x}} < c(\mathbf{x} \parallel \bar{x}) + \alpha + (\gamma - 2\alpha)/2.$$

From Lemma 5.6, we have

$$c_x > \gamma - \alpha - (\gamma - 2\alpha)/2 = \gamma/2, \ c_{\bar{x}} < 0 + \alpha + (\gamma - 2\alpha)/2 = \gamma/2.$$

and hence $c_x > c_{\bar{x}}$. This implies that the lines 6-9 of Algorithm 3 will set $\mathbf{x} \leftarrow \mathbf{x} \parallel x$. By definition of $x$, the invariant that $\mathbf{x} \sqsubseteq \mathbf{z}$ is preserved.

Therefore, the algorithm solves $\mathsf{FindZ}_{\ell,\gamma,\varepsilon}(\widetilde{G}_{\mathbf{z}}^{\alpha})$.

$\square$

**Lemma B.1.** *For any $\tau > 0$, $\beta > 0$, $c^{\alpha}(\mathbf{x}) = \Pr[\widetilde{G}_{\mathbf{z}}^{\alpha}(S, \mathbf{x})$ credits $s_1]$ can be estimated within additive error $\tau$, with probability $1 - \beta$, using $O(\tau^{-2} \log(2/\beta))$ oracle queries from $\widetilde{G}_{\mathbf{z}}^{\alpha}(S, \mathbf{x})$.*

*Proof of Lemma B.1.* Independently sample $N = \tau^{-2} \log(2/\beta)$ outcomes $(y_1, C_2), \ldots, (y_N, C_N)$ on input $(S, \mathbf{x})$ from the output distribution of $\widetilde{G}_{\mathbf{z}}^{\alpha}(S, \mathbf{x})$ by making queries to $\widetilde{G}_{\mathbf{z}}^{\alpha}$. Let $X_i = \mathbf{1}[s_1 \in C_i]$ and we have $\mathbb{E}[X_i] = \mathbb{E}[\mathbf{1}[s_1 \in C_i]] = \Pr_{(x,C_i) \sim \widetilde{G}_{\mathbf{z}}^{\alpha}(S,\mathbf{x})}[s_1 \in C_i] = c^{\alpha}(\mathbf{x})$. Since $X_i$s are independent, by Chernoff-Hoeffding inequality, we have

$$\Pr\left[\left|\frac{1}{N}\sum_{i=1}^{N} X_i - c^{\alpha}(\mathbf{x})\right| \geq \tau\right] \leq 2\exp\left(\frac{-2\tau^2}{N \cdot (\frac{1}{N} \cdot 1)^2}\right) = \beta.$$

$\square$

