# OpenReview forum: "Barriers to Counterfactual Credit Attribution for Autoregressive Models"
_ICML.cc/2026/Conference — ICML 2026 regular_

### Official Review · Reviewer_Rj6E · 2026-03-06

**Soundness:** 3
**Presentation:** 3
**Significance:** 2
**Originality:** 3
**Overall Recommendation:** 4
**Confidence:** 2

**Summary:**

This paper initiates the study of Counterfactual Credit Attribution (CCA) in the context of autoregressive language models. The authors investigate two natural approaches to building CCA generative models and uncover fundamental barriers for both. First, they show that CCA does not compose under autoregressive generation: even a (0,0)-CCA next-token predictor, which is the strongest possible, can yield a rollout that is not (ε,δ)-CCA for any ε≥0 and δ<1, with a quantitative lower bound on the composition degradation. Second, they study CCA retrofitting, which wraps a non-crediting model with a black-box algorithm that adds credit sets, and prove that any such algorithm requires Ω(2^ℓ) queries to the base model in the worst case, where ℓ is the output length. This holds even for α-approximate solutions with α < 1/2. The proofs rely on constructing a hard family of next-token predictors indexed by binary strings, where finding the hidden string needed for optimal credit attribution requires exponentially many queries.

**Compliance With Llm Reviewing Policy:**

Affirmed.

**Final Justification:**

No discussion, maintain my score.

**Key Questions For Authors:**

see weaknesses

**Limitations:**

yes

**Strengths And Weaknesses:**

Strengths
- This is the first paper to formally study CCA in the context of autoregressive generative models. The problem is timely and important, as generative AI is increasingly deployed with retrieval-augmented generation (RAG) and in-context learning, the question of properly attributing credit to data sources has significant legal, ethical, and practical implications.
- The theoretical results appear to be technically correct and rigorous. The non-composition result is elegant, the counter example is simple yet the conclusion is surprising. The lower bound for CCA retrofitting is well-constructed: the hard model family is clever, and the proof strategy of reducing the retrofitting problem to a hidden string identification problem is clean and convincing.

Weaknesses
- The paper is purely theoretical with no empirical component. While this is acceptable for a theory paper, the hard instances constructed (binary token vocabulary, singleton data universe S = {s₁}) are far from practical LLM settings. It remains unclear whether these barriers manifest in practice for real language models with large vocabularies and datasets. The paper would benefit from even a small-scale empirical illustration showing the composition degradation or retrofitting difficulty on a simple but more realistic model.
- Theorem 4.3 shows a counterintuitive result: the lower bound on ε' of the rollout grows as the next-token predictor's ε → 0 (i.e., better token-level attribution leads to worse composition). However, the authors caution that ε is endogenous to the model, making interpretation difficult. This complicates the takeaway message and is not fully resolved.

---

> ### Author Rebuttal · Authors · 2026-03-28
>
> Thank you for your thoughtful comments.
>
> ## Answers to questions
> 1. We agree that it would be nice to supplement the theory with experiments.
> 2. We acknowledge the limitations of this theorem as explained in the paper.

---

### Official Review · Reviewer_XvSf · 2026-03-16

**Soundness:** 4
**Presentation:** 4
**Significance:** 4
**Originality:** 3
**Overall Recommendation:** 4
**Confidence:** 4

**Summary:**

This paper studies credit attribution for generative models. The authors focus on Counterfactual Credit Attribution (CCA), a technical condition inspired by differential privacy and recently introduced by Livni et al. (2024), and examine how it applies to autoregressive language models during deployment, such as when using a retrieval-augmented generation (RAG) database. Roughly speaking, this notion requires that for any fixed prompt $x$, if the algorithm does not credit a source $s_i$, then the output distribution of the algorithm when $s_i$ is dropped is $(\varepsilon,\delta)$-close to the original one.

The authors identify significant barriers to two natural methods for achieving CCA. First, they prove that CCA does not compose autoregressively. This means that even if an underlying next-token predictor perfectly satisfies the CCA condition (for outputting the next token), the resulting sequence of generated tokens is not guaranteed to satisfy it. This is in sharp contrast to differential privacy.

Second, the paper demonstrates that "retrofitting" is computationally infeasible. Retrofitting involves taking an existing, non-crediting model in a black-box manner, and adding a wrapper to calculate and assign credit. The authors show that achieving a CCA-compliant model this way requires a number of queries that scales exponentially with the length of the model's output.

**Compliance With Llm Reviewing Policy:**

Affirmed.

**Key Questions For Authors:**

Currently, the authors propose a definition that takes en ex-ante view of credit attribution: the generative model takes as input a prompt $x$ and a set of documents $S$ and forms a distribution $\Delta_{x,S}$ over responses $y$. The definition requires that if we drop an input $i$ that is not credited, then the resulting distribution $\Delta_{x,S_{-i}}$ is $(\epsilon,\delta)$-close to the original one. I'm not sure that the ex-ante view is the correct one for credit attribution; instead one could imagine a definition that has an ex-post flavor and says that the credit should be based on the *sampled* output $y$ and allow the set of credited sources to depend on $y$. The definition can have a similar flavor and ask that if some source $i$ isn't credited when $y$ is sampled, then the probability of sampling $y$ under $S_{-i}$ should be $(\epsilon,\delta)$-close to the original one. If one were to draw a (rough) analogy with how attribution is done in the academic literature, the list of credit sources depends on the *realized* paper that some researcher writes, not on the *distribution* over potential papers it can write. Is there a reason to support the ex-ante view you propose in your paper instead of the ex-post view? The ex-post attribution is also much easier to audit given access to the log-likelihoods of a generative model.

**Limitations:**

Yes.

**Strengths And Weaknesses:**

Strengths:
- The paper studies an interesting and timely problem.
- The paper is well-written and easy to follow.
- The results have interesting conceptual takeaways.

Weaknesses:
- I'm not fully convinced that this type of CCA definition is the correct way to be thinking about credit attribution. Please see my question below.
- It would have been nicer if the authors had studied relaxations of this definition (as they suggest in the paper) and seen if they can remedy the negative results they have shown.

Overall, despite the fact that I'm not fully convinced about the definition, I'm currently leaning towards "weak accept" since I think it is useful for the community to think of the correct definitions in that space. I might revise based on the rebuttal discussions.

---

> ### Author Rebuttal · Authors · 2026-03-28
>
> Thank you for your thoughtful comments and for the insightful question. We are glad that you find important the question of seeking the right definitions.
>
> ## Answers to questions
> The ex-post variant of CCA seems interesting and perhaps easier to achieve (to the extent that we understand the definition you have in mind). We do not claim that CCA is the final word on defining credit attribution (eg, Remark 5.6.1). Different definitions may be more or less appropriate in different settings, or we might discover that there is an altogether better paradigm. The results of this paper also inform the community about the properties and potential limitations of CCA.
>
> In the course of our research, we have also considered multiple variations on the CCA definition. However, for the scope of **this** paper, we focus instead on applying CCA as-is to generative models.

---

> > ### Author Rebuttal · Reviewer_XvSf · 2026-04-03
> >
> > Thanks for your comment. I remain positive about the paper.

---

### Official Review · Reviewer_NCVe · 2026-03-17

**Soundness:** 3
**Presentation:** 2
**Significance:** 2
**Originality:** 3
**Overall Recommendation:** 4
**Confidence:** 2

**Summary:**

The work  investigates the theoretical feasibility of applying Counterfactual Credit Attribution (CCA) -- a DP-inspired framework -- to autoregressive generative models during deployment. It identifies two major mathematical barriers to building CCA-compliant LMs. First, the authors prove that CCA does not compose autoregressively, meaning a CCA-compliant next-token predictor can yield a non-compliant sequence. Second, they show that retrofitting an existing model to satisfy CCA via black-box API access is computationally intractable and requires a number of queries exponential to the output sequence length. To summarize, the work highlights that achieving instance-level data attribution in generative models likely requires white-box interventions or relaxed definitions rather than post-hoc wrappers.

**Compliance With Llm Reviewing Policy:**

Affirmed.

**Key Questions For Authors:**

- In def. 5.1, the authors require that the retrofitted model $\tilde{G}$ perfectly preserve the marginal distribution over the generated text of the base model $G$. This is a strong constraint. Instead of requiring the distributions to match exactly, we can could allow the retrofitted model $\tilde{G}$ to deviate from the base model $G$ by a small tolerance $\eta$, e.g., measured via Total Variation (TV) distance or Kullback-Leibler (KL) divergence. Could an approach along these lines to relax this constraint in def. 5.1 and achieve polynomial-time retrofitting?

**Limitations:**

yes

**Strengths And Weaknesses:**

**Strenghts**

- The work studies a timely topic. It uses the concept of Counterfactual Credit Attribution (CCA) -- which was designed for basic PAC learning -- and formalizes it for the autoregressive generative models we use (e.g., in RAG systems or in-context learning). Essentially they show for autoregressive LMs the concept of CCA breaks, demonstrating CCA's limitations for credit attribution in LMs.

- The theoretical analysis of the two most intuitive approaches to attribution are interesting and relevant despite the limited edge cases (i.e., $\delta=0). First, the authors show taht CCA does not compose autoregressively (thm 4.2) and second they show that black-box retrofitting requires exponential queries (thm 5.5).

**Weaknesses**

- The exponential query lower bound in thm 5.5 relies on two strong assumptions: pure CCA (where parameter $\delta$ is exactly 0) and 'exact augmentation' (i.e., the retrofitted model cannot change the original text generation distribution). In practical DP, we almost always relax these constraints (e.g., using approximate DP or allowing slight distribution shifts) to achieve tractability.

- The work would benefit from an empirical section to show the gap between worst-case bounds and what is done in practice on an average-case basis. For example, it woudl be valuable to see how quickly the $\epsilon'$ parameter degrades during autoregressive rollout of a small LLM (e.g., a mini LLaMA or GPT-2). This may show that composition failure in Theorem 4.3 is a practical problem and not only a mathematical edge case.

- Minor: By focusing on deployment-time datasets (e.g., (system) prompts or RAG databases), the paper goes arround the much larger issue of attributing credit to large datasets used during pre-training. While deployment-time attribution is also an interesting problem, leaving out training-time attribution makes the copyright/IP motivation from the intro a little bit disconnected from the actual contribution of this work.

---

> ### Author Rebuttal · Authors · 2026-03-28
>
> Thank you for your thoughtful comments. We agree that it would be nice to supplement the theory with experiments. The other directions you identified as Weaknesses are natural directions for future work, as discussed in Section 6 and Line 193.
>
> One small point we would like to clarify is that the autoregressive non-composability result (Thm 4.2) is not limited to $\delta=0$.
>
> ## Answers to questions
> We posed the same question in Section 6 (Discussion and Open Question). Trivially, when the allowable tolerance $\eta$ on the distribution change is large enough, one can simply make the marginal distribution of $\tilde{G}$ with a document $s$ on each prompt be the same as that without the document. In this case, always giving no credit already satisfies CCA and is optimal. However, when $\eta$ is very small but non-zero, we conjecture that there will be some $\bar{\alpha}(\eta) < 1/2$ such that the exponential lowerbound still holds for any $\alpha$-additive approximate optimal with $\alpha < \bar{\alpha}(\eta)$.

---

> > ### Author Rebuttal · Reviewer_NCVe · 2026-04-06
> >
> > My questions have been addressed. I will keep the current score.

---

### Official Review · Reviewer_W413 · 2026-03-24

**Soundness:** 4
**Presentation:** 3
**Significance:** 3
**Originality:** 4
**Overall Recommendation:** 5
**Confidence:** 3

**Summary:**

The authors study the feasibility of applying the concept of counterfactual credit attribution (CCA) to autoregressive models (in the context of generative models). The focus is on deployment-time datasets (such as RAG and ICL). The authors study the difficulties in applying CCA to generative models and propose two direct approaches to apply CCA to generative models. Their theoretical results show that both the methods are infeasible in the context of autoaggressive models (a) CCA cannot compose autoregressively unlike pure DP and (b) retrofitting is computationally infeasible in the worst case for black box access to models. The authors also discuss open questions in applying CCA to generative models.

**Compliance With Llm Reviewing Policy:**

Affirmed.

**Key Questions For Authors:**

Please see Weakness 1 and Comment 2

**Limitations:**

Yes

**Strengths And Weaknesses:**

Strengths:

1. The authors study the (first time) application of CCA in the context of generative models.
2. The authors prove that two natural approaches (with given conditions) cannot be applied in theory thus driving the relatively new field of CCA for generative models.
3. The authors discuss open questions and possibilities of applying CCA to generative models that drives the theoretical research on CCA similar to DP.

Weaknesses:

1.	Discussion comparing other data attribution methods and CCA in terms of feasibility and advantages. Why should one go for CCA (discuss Line 386 under Remark 5.6.1), if approximate methods exist for attribution? Can’t the Shapely values be turned to binary?
2.	The strength of the paper is its negative theoretical results. Adding simple empirical experiments such as an approximate retrofitter to study the gap could be included in Section 6 (maybe appendix) that would be more encouraging to pursue open questions and understand the gap between theory and practice, given CCA is relatively new. Or more discussion around one of the open questions, for example, credit-optimal CCA augmentation to drive future work. This would add encouraging positive results to the paper.

Comments:

1.	Typo on Line 154, is it $\mathbf{x}= \mathbf{x} || x$?
2.	It would be helpful for readers, if $\epsilon$ is explained wrt CCA in preliminaries. In DP high $\epsilon$ means higher privacy risk. In CCA, what does the value of $\epsilon$ imply, example: $\epsilon = 0$ imply that no documents need to be credited?

---

> ### Author Rebuttal · Authors · 2026-03-28
>
> Thank you for your thoughtful comments and for the typo. We agree that it would be nice to supplement the theory with experiments.
>
> ## Answers to questions
>
> ### Weakness 1
> CCA is trying to capture different aspects of the data attribution problem compared to the Shapley value. CCA was originally proposed as an answer to the question: What property ``should'' a binary data attribution satisfy?
>
> Shapley value is the answer to a different question: It is the unique real value assigned to each data point satisfying certain desirable fairness axioms given the contribution of the data points. It is true that rounding Shapley value (or other real-valued measures) to binary values would result in some binary data attribution method, but it is not clear what desirable properties the result would satisfy. However, rounding the Shapley value to binary outcomes would break the original axioms that motivate Shapley in the first place. Therefore, how one should round real-valued data contribution measures to binary data attribution is a good question for future work.
>
> As a final technical point, using Shapley value requires selecting an utility / loss function.  CCA does not depend on specific optimization objectives.
>
> ### Comment 2
> $\epsilon$ captures the looseness of the attribution guarantee. Higher $\epsilon$ implies greater tolerance for letting a document influence the output distribution without giving it credit. $\epsilon=0$ implies that if a document is not given credit, then it can not influence the output distribution at all. On the other extreme, $\epsilon \to \infty$ implies that even if a document is not given credit, it can have a large effect on the output distribution.

---

> > ### Author Rebuttal · Reviewer_W413 · 2026-04-01
> >
> > My questions have been addressed. I will keep the current score.

---

### Decision · Program_Chairs · 2026-04-30

**Decision:**

Accept (regular)

**Comment:**

The reviewers agreed that this paper studies a timely problem and makes technically strong theoretical contributions by formalizing CCA for autoregressive generative models and identifying fundamental barriers to two natural approaches. Multiple reviewers highlighted the rigor, originality, and conceptual value of the non-composition and retrofitting impossibility results, and the rebuttal adequately addressed the main clarification questions raised during review.

Based on the technical soundness, originality, and likely value of these results to the emerging literature on attribution for generative models, I recommend acceptance.